# Voltage-driven gigahertz frequency tuning of spin Hall nano-oscillators

Jong-Guk Choi[1,4], Jaehyeon Park[2,4], Min-Gu Kang[1], Doyoon Kim[3], Jae-Sung Rieh[3], Kyung-Jin Lee [2], Kab-Jin Kim [2✉] & Byong-Guk Park [1✉]

Spin Hall nano-oscillators (SHNOs) exploiting current-driven magnetization auto-oscillation have recently received much attention because of their potential for neuromorphic computing. Widespread applications of neuromorphic devices with SHNOs require an energy-efficient method of tuning oscillation frequency over broad ranges and storing trained frequencies in SHNOs without the need for additional memory circuitry. While the voltage-driven frequency tuning of SHNOs has been demonstrated, it was volatile and limited to megahertz ranges. Here, we show that the frequency of SHNOs is controlled up to 2.1 GHz by an electric field of 1.25 MV/cm. The large frequency tuning is attributed to the voltage-controlled magnetic anisotropy (VCMA) in a perpendicularly magnetized Ta/Pt/[Co/Ni]$_n$/Co/AlO$_x$ structure. Moreover, the non-volatile VCMA effect enables cumulative control of the frequency using repetitive voltage pulses which mimic the potentiation and depression functions of biological synapses. Our results suggest that the voltage-driven frequency tuning of SHNOs facilitates the development of energy-efficient neuromorphic devices.

---

[1] Department of Materials Science and Engineering, KAIST, Daejeon 34141, Korea. [2] Department of Physics, KAIST, Daejeon 34141, Korea. [3] School of Electrical Engineering, Korea University, Seoul 02841, Korea. [4]These authors contributed equally: Jong-Guk Choi, Jaehyeon Park. ✉email: kabjin@kaist.ac.kr; bgpark@kaist.ac.kr

Spintronic oscillators that generate microwaves through spin-torque-driven magnetization precession are being extensively investigated for neuromorphic applications[1–6] owing to their unique features, including non-linearity[7], low-power operation[8,9], scalability[1,4,10], and CMOS compatibility[4,11]. Indeed, spin-transfer-torque-based oscillators (STOs) have recently been implemented in oscillator neural networks which demonstrate pattern recognition[2,12,13]. The STO device has been successfully trained to recognize the input pattern signals by tuning the frequency of individual oscillators using a driving current[2,12] or magnetic field[13]. However, since the current or magnetic field-based frequency tuning is not energy-efficient, it is necessary to find an alternative way to widely tune the frequency with reduced power consumption.

Recently, another type of STO has been developed in which magnetization precession is caused by spin currents generated by the spin Hall effect[14,15]. This so-called spin Hall nano-oscillator (SHNO) is operated with an in-plane current[16–20], and has the following advantages over the conventional STO operated with a perpendicular current. First, multiple oscillators integrated into a common spin-Hall material can be controlled simultaneously by an in-plane current, allowing long-range mutual synchronization of the SHNOs[21–23]. Second, a gate structure is easily incorporated in a three-terminal SHNO structure[24–26], allowing an individual oscillator to be independently controlled by the gate voltage[27,28].

In a recent study[28], the gate voltage-induced frequency tuning in W/CoFeB/MgO structures with a frequency tuning range of ~50 MHz was experimentally demonstrated. Compared to the oscillation frequency (~10 GHz), this frequency tuning range is relatively narrow. Although 10~50 MHz frequency tunability is sufficient to optimize the synchronization map for a small number of different synchronization state as in vowel recognition[2,28], a wider frequency tunability is required for tasks that require more states to be recognized. This would expand the applicability of gate-controlled SHNOs to widespread oscillator-based neuromorphic computing. Moreover, since the previously demonstrated voltage-driven frequency tuning of SHNOs[28] is volatile, additional memory circuitry is required to store trained synaptic weights of oscillation frequency.

In this work, we report the voltage-driven large frequency tuning of SHNOs by exploiting voltage-controlled magnetic anisotropy (VCMA)[29–34], as schematically illustrated in Fig. 1a. The SHNO is composed of a ferromagnet (FM)/heavy metal (HM) bilayer. In this geometry, a charge current flowing through the HM layer generates a vertical spin current via the spin Hall effect, which exerts spin-orbit torques (SOTs) on the magnetization of the FM layer. The auto-oscillation of magnetization occurs when the SOT (green arrow) fully compensates the damping torque (gray arrow), and the auto-oscillation frequency is governed by the resonance frequency ($f_{res}$) of the FM. Here, we employ a perpendicular magnetic anisotropy (PMA) material, which is distinct from the in-plane magnetized materials generally used in previous works. In the presence of an in-plane field, $f_{res}$ of a PMA material is determined as[35–37],

$$f_{res} = \frac{\gamma}{2\pi}\sqrt{B_k^2 - B_\parallel^2}\left(0 < B_\parallel < B_k\right),$$

$$f_{res} = \frac{\gamma}{2\pi}\sqrt{B_\parallel\left(B_\parallel - B_k\right)}\left(B_\parallel \geq B_k\right), \quad (1)$$

where $\gamma$ is the gyromagnetic ratio, $B_k$ is the effective perpendicular anisotropy field, and $B_\parallel$ is the external in-plane magnetic field. Equation (1) indicates that a large frequency tuning is effectively achieved by controlling $B_k$. Figure 1b shows an example; when $B_\parallel$ is larger than $B_k$, the frequency increases by weakening $B_k$. In this work, we demonstrate that the $B_k$ of

perpendicularly magnetized Co/Ni multilayers is controlled at values as large as 0.22 T due to a moderate electric field of 1.25 MV/cm. This results in frequency tuning up to 2.1 GHz, which is more than 40 times larger than previously reported values[27,28]. We find that voltage-controlled frequency tuning is independent of the driving current and is significantly more efficient than the current-controlled frequency tuning used in conventional STOs. Furthermore, the voltage effect is non-volatile in our sample, allowing the cumulative frequency control of SHNOs using repetitive voltage pulses. This can facilitate the potentiation and depression functions of artificial synapses[38–41], and thus can be utilized in the learning process of neuromorphic devices. Our results demonstrate the efficient frequency tuning of SHNOs via gate voltage provides an important building block for spin-based low-power neuromorphic hardware applications.

## Results

**Voltage-controlled perpendicular magnetic anisotropy.** We first demonstrate the VCMA effect in a Co/Ni multilayer sample of Ta (3 nm)/Pt(5 nm)/[Co(0.45 nm)/Ni(0.6 nm)]$_7$/Co(0.45 nm)/AlO$_x$ (2 nm) structures (see Methods for details of sample growth). Figure 1c shows the magnetization measurement while sweeping a perpendicular magnetic field ($B_z$), indicating that the Co/Ni film has PMA. To test the VCMA of the sample, we patterned the film into a Hall bar device with a 10 μm × 10 μm cross, which is fully covered by a gate oxide of ZrO$_2$ (40 nm) and a gate electrode of Ru (50 nm). Figure 1d shows the anomalous Hall resistance ($R_H$) after sequentially applying a gate voltage ($V_g$). Here, we applied a $V_g$ to the top electrode for 5 minutes at 150 °C and then measured $R_H$ at room temperature. This is possible because the voltage effect is maintained even after turning off the $V_g$[32,33,42,43] (Supplementary Note 1). The result shows that the PMA is gradually weakened by positive $V_g$'s and restored by subsequent negative $V_g$'s, indicating that the PMA in our Co/Ni multilayer sample is effectively modulated by $V_g$. Note that the $V_g$ of 5 V is equivalent to an electric field of 1.25 MV/cm. We note that the VCMA effect can be improved by materials engineering as it has been demonstrated that the gate effect occurs at room temperature and short voltage pulses by introducing a thin or double gate oxide (Supplementary Note 2)[44].

In order to quantitatively analyze the VCMA effect, we performed the spin-torque-ferromagnetic resonance (ST-FMR) measurement. The ST-FMR spectra were measured using a bar-shaped device (8 μm × 14 μm) fabricated from the same Co/Ni film. The microwave frequency used for the ST-FMR measurement ranges from 10 GHz to 21 GHz (see Methods for details of measurement). Figure 2a shows the ST-FMR spectra of the sample with different values of $V_g$ at a frequency of 15 GHz, which were measured while sweeping magnetic fields in a direction slightly tilted from the z-direction. Note that the out-of-plane magnetic field geometry was employed to accurately determine the PMA, avoiding artifacts that might arise in the in-plane magnetic field geometry, such as local anisotropy variation or two magnon scattering[45] (Supplementary Note 3). The ST-FMR spectra can be expressed by the combination of symmetric and anti-symmetric Lorentzian functions as[46,47],

$$V_{ST-FMR}(B_z) = V_{sym}\frac{\triangle B^2}{\left(B_z - B_{res,z}\right)^2 + \triangle B^2} + V_{asy}\frac{\triangle B(B_z - B_{res,z})}{\left(B_z - B_{res,z}\right)^2 + \triangle B^2}, \quad (2)$$

where $B_{res,z}$ is the center magnetic field of the Lorentzian functions corresponding the resonance magnetic field, $\triangle B$ is the linewidth (half width at half maximum) of the Lorentzian function, and $V_{sym}(V_{asy})$ is symmetric (anti-symmetric) component of resonance amplitude. The dotted lines represent the fitting curves of the

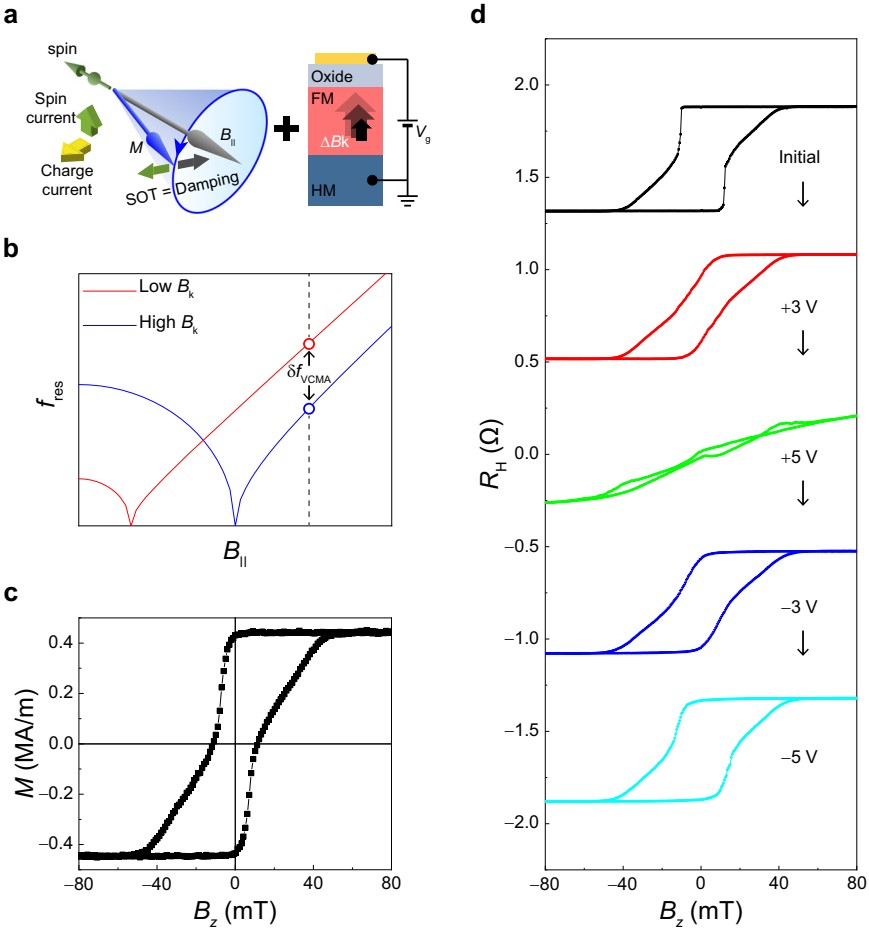

**Fig. 1 Voltage-controlled perpendicular magnetic anisotropy. a** Schematic illustration of the mechanism of the voltage-driven frequency tuning via voltage-controlled magnetic anisotropy (VCMA). The green and gray arrows represent the directions of spin-orbit torque and damping torque, respectively. **b** The resonance frequency ($f_{res}$) as a function of in-plane magnetic field ($B_{||}$) for high PMA (blue) and low PMA (red). **c** Magnetization ($M$) versus out-of-plane magnetic field ($B_z$) of the Co/Ni film. **d** Anomalous Hall resistance ($R_H$) curves of the Co/Ni sample for sequentially applied gate voltages of +3 V, +5 V, −3 V, and −5 V, respectively.

ST-FMR spectra using Eq. (2), from which we extracted the $B_{res,z}$ and $\triangle B$ values. It is observed that the Lorentzian curve for the initial sample (black symbols) is centered around 0.34 T. That is, $B_{res,z} = 0.34$ T. Notably, the $B_{res,z}$ is largely shifted by the $V_g$; the $B_{res,z}$ is increased up to 0.59 T by a positive voltage (+5 V, red symbols), but it is then reduced to 0.39 T by a subsequent negative voltage (−5 V, blue symbols). We measured the $B_{res,z}$ for different frequencies (Supplementary Note 4) and summarize the resonance frequency $f_{res,z}$ vs $B_{res,z}$ in Fig. 2b, where the black symbols represent the initial sample without applying a $V_g$, while the red and blue symbols correspond to the samples after successively applying +5 V and −5V, respectively. The linear relation between the $f_{res,z}$ and $B_{res,z}$ is well explained by the Kittel formula $f_{res,z} = \frac{\gamma}{2\pi}(B_{res,z} + B_k)$[45], from which the $B_k$ can be extracted. We note that the ST-FMR measurement was done under a magnetic field along the z-direction, so above resonance frequency formula is different from Eq. (1), which is based on an in-plane magnetic field. Figure 2c shows the extracted $B_k$ according to the $V_g$: the $B_k$ reduces from $0.16 \pm 0.01$ T to $−0.08 \pm 0.01$ T when applying +5 V, and is restored close to the initial value by a negative gate of −5 V. This is consistent with the results shown in Fig. 1c. Note that the average change of $B_k$ is $0.22 \pm 0.02$ T by ±5 V, which can cause a frequency tuning up to a few GHz according to Eq. (1). Note that the sample resistance increases (decreases) at negative (positive)

voltages, suggesting that oxidation or reduction occurs at the ferromagnet/oxide interface when voltage is applied. In addition, a sample whose oxidation level is intentionally reduced (increased) by controlling the oxidation time has a low (high) PMA, which is the same effect as applying a positive (negative) gate voltage (Supplementary Note 4). This indicates that the VCMA effect in our samples is dominated by the voltage-driven oxygen ion migration.

Figure 2d shows the $\triangle B$ as a function of $f_{res,z}$, where the color symbols represent the samples with different values of $V_g$, identical to those in Fig. 2b. Using the relation $\triangle B = \triangle B_0 + \frac{2\pi\alpha_{eff}}{\gamma}f_{res,z}$[45–48], the effective magnetic damping constant $\alpha_{eff}$ is obtained. Here, $\triangle B_0$ is the zero-frequency linewidth due to long-range magnetic inhomogeneity[48]. Fig. 2f shows the dependence of the $\alpha_{eff}$ on $V_g$; the $\alpha_{eff}$ decreases (increases) by a positive (negative) $V_g$. The change in $\alpha_{eff}$ by gate voltage is also attributed to the voltage-induced modification of the interfacial oxidation state (Supplementary Note 5), possibly through the modulation of interfacial Rashba spin-orbit-coupling (RSOC)[49] as the surface (or interface) oxidation increases the RSOC[49] and the RSOC modifies the damping[50,51].

**Voltage-driven frequency tuning of an SHNO.** We now move on to our main result, the gate voltage-induced frequency tuning

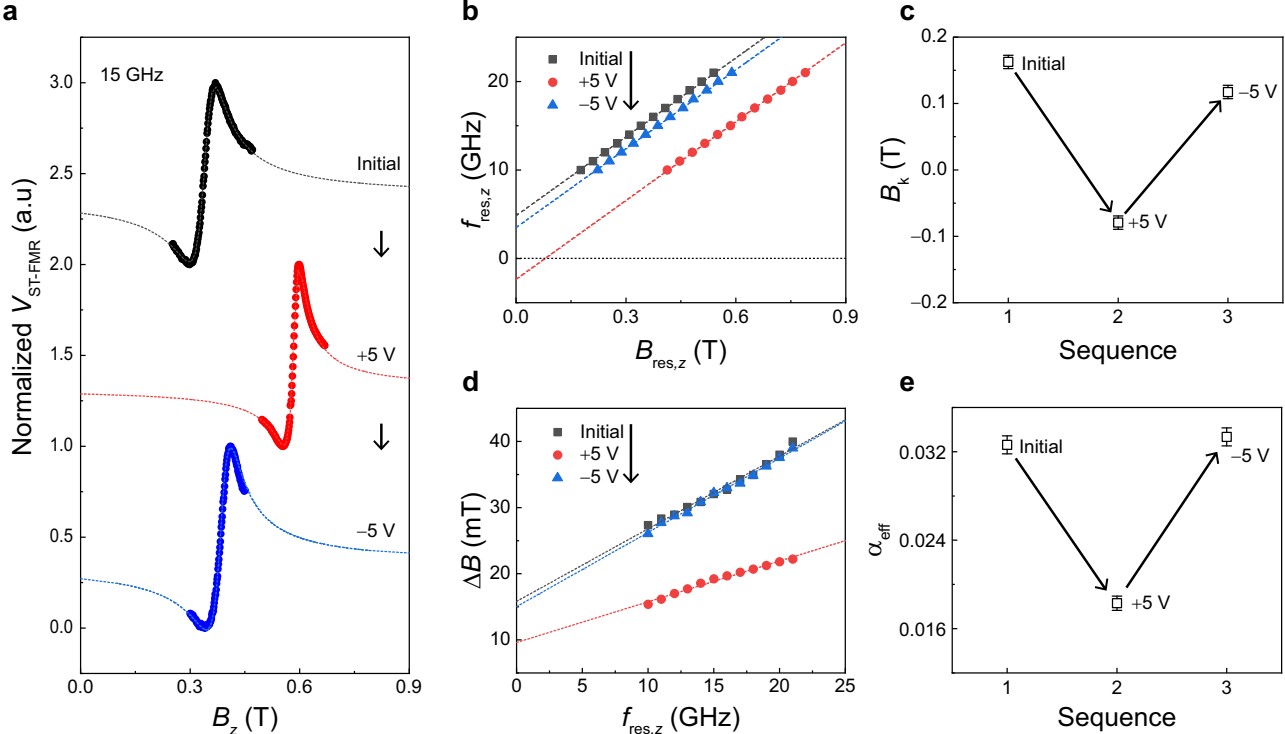

**Fig. 2 Voltage-dependent ST-FMR spectra. a** ST-FMR spectra of the Co/Ni sample for sequentially applied gate voltages (initial, +5 V, and −5 V). Here, the microwave frequency is 15 GHz. The dotted lines are the best fits based on Eq. (1). **b** Resonance frequency ($f_{res,z}$) as a function of resonance field $B_{res,z}$ of the sample with different sequentially applied gate voltages. **c** Variation of perpendicular magnetic anisotropy field ($B_k$) versus sequentially applied gate voltage, where the error bars are due to the uncertainty in fitting the data in Fig. 2b to the Kittel formula, $f_{res,z} = \frac{\gamma}{2\pi}(B_{res,z} + B_k)$. **d** The linewidth of the Lorentzian function ($\triangle B$) as a function of the $f_{res,z}$ of the sample with sequentially applied gate voltages. **e** Variation of effective damping constant ($\alpha_{eff}$) versus sequentially applied gate voltage, where the error bars are due to the uncertainty in fitting the data in Fig. 2d to the equation, $\triangle B = \triangle B_0 + \frac{2\pi\alpha_{eff}}{\gamma}f_{res,z}$.

of an SHNO. Figure 3a includes an illustration of the measurement schematics; the SHNO device was fabricated by patterning the Co/Ni film into a nano-constriction with a width of 140 nm, which is entirely covered by a gate oxide of $ZrO_2$ (40 nm) and a gate electrode of Ru (50 nm). A dc current ($I_{dc}$) is applied along the constriction channel (x-direction) under a magnetic field (B) applied in the direction with a polar angle θ of 80° and an azimuthal angle φ of 70°. Here, we applied a magnetic field along the nearly in-plane direction to maximize the SOT-induced anti-damping effect. The slight tilt from the plane suppresses the SOT-induced magnetization switching[52]. The azimuthal angle was chosen to achieve a large electrical auto-oscillation signal[17,18]. In this geometry, the $I_{dc}$ generates SOTs through the spin Hall effect in the Pt underlayer, which leads to a magnetization oscillation of the Co/Ni layer when the SOT compensates for the damping torque. The oscillation in magnetization causes periodic change in the MR of the same frequency[11], which is detected by a power spectral density (PSD) (see Methods for more details of measurement). To check whether the MR of the nano-constricted device is large enough to generate a detectable PSD, we measured the MR of the nano-constriction sample using an in-plane rotating magnetic field of 1 T (Fig. 3b). The MR ratio is ~1.2%, which is sufficient to monitor the auto-oscillation of magnetization[17,19,20].

Figure 3c shows the color plots of PSD as a function of a magnetic field (B) for the initial sample without applying a $V_g$. Here, we used a fixed $I_{dc}$ of 2.9 mA. The auto-oscillation peak is clearly visible and the peak frequency increases with increasing magnetic field, demonstrating that our Co/Ni device successfully operates as an SHNO. We then investigate the effect of $V_g$ on the auto-oscillation. The $V_g$ was applied in the same manner as the

$R_H$ measurement shown in Fig. 1. Figures 3d–g show the results, where the $V_g$'s of +4 V, +5 V, −2 V, and −3 V were successively applied. Notably, the auto-oscillation frequency is vertically shifted by the $V_g$'s; it increases (decreases) with a positive (negative) $V_g$. These results are consistent with our expectation, as illustrated in Fig. 1a, that a positive (negative) $V_g$ reduces (enhances) PMA or $B_k$ and consequently increases (decreases) the oscillation frequency. To clearly visualize the gate voltage-induced frequency tuning, we extract the auto-oscillation spectra for B = 0.56 T from Figs. 3c–g and plot them in Fig. 3h. This demonstrates that frequency tuning takes place up to 2.1 GHz with a $V_g$ of 5 V, which is remarkably larger than the previously reported value of 50 MHz[27,28] with a similar $V_g$. This wider frequency tunability allows more states to be recognized, expanding the applicability to a wide range of oscillator-based neuromorphic computing. Note that similar results are also observed in another Co/Ni device with a different thickness (Supplementary Note 6), confirming the reproducibility of the voltage-controlled frequency tuning of the SHNO.

We next investigate the current-dependent auto-oscillation of the SHNO. Figures 4a–e show PSD spectra as a function of current for the sample with different values of $V_g$. Here, we used a fixed magnetic field of 0.56 T. It is found that the application of the $V_g$ changes two properties of the SHNO: the auto-oscillation frequency and the threshold current ($I_{th}$) at which auto-oscillation begins to occur. First, the auto-oscillation peaks shift up and down according to the $V_g$, demonstrating voltage-controlled frequency tuning, consistent with the results shown in Fig. 3. Note that the slight increase in oscillation frequency with increasing $I_{dc}$ is attributed to the increase in the SOT[7,53,54].

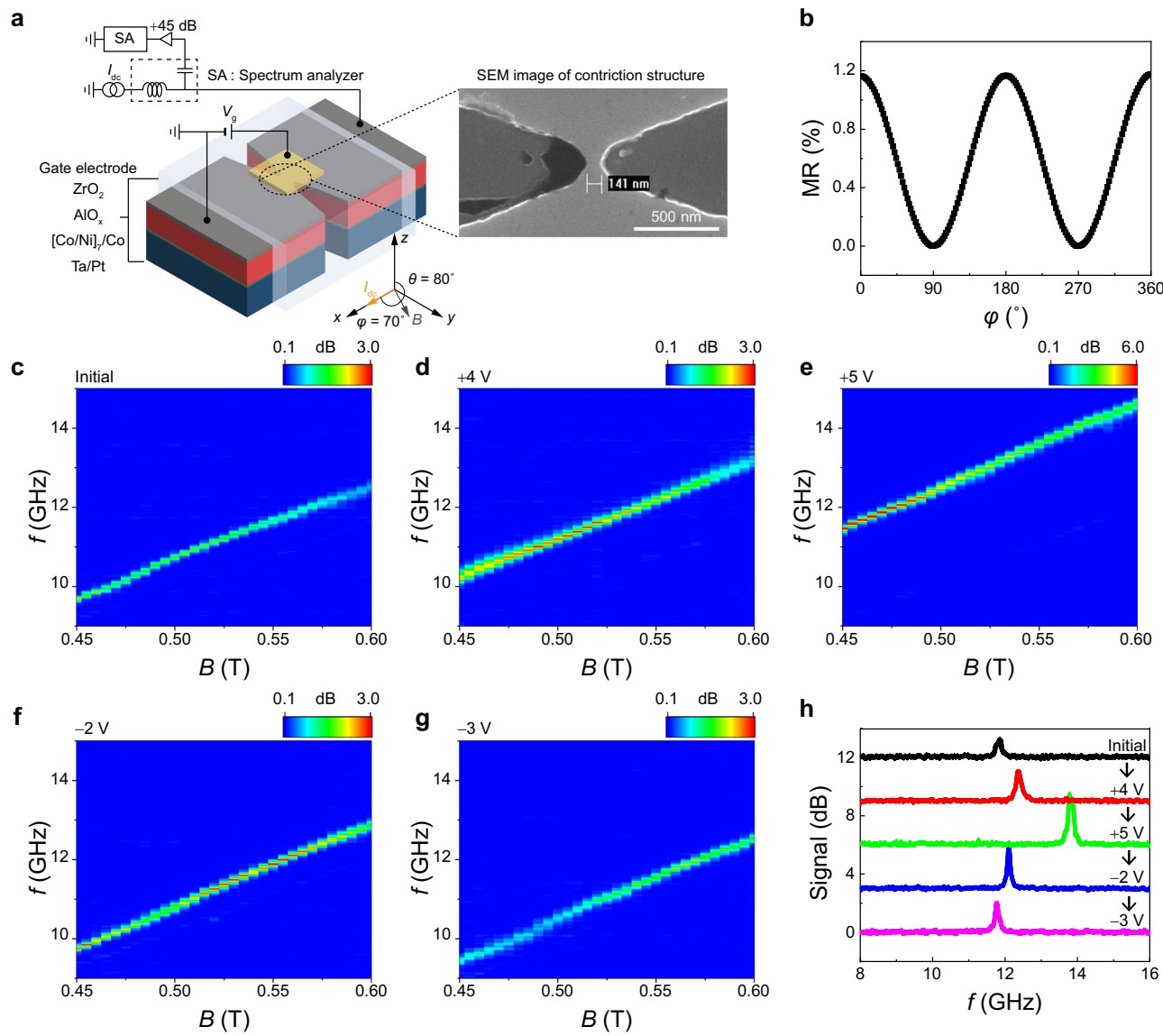

**Fig. 3 Magnetic field-dependent voltage-driven frequency tuning of SHNO. a** Schematic illustration of the experimental set-up. The inset is a scanning electron microscope image of a SHNO device. **b** Angle-dependent magnetoresistance (MR) of the Co/Ni sample. **c–g** Power spectral densities (PSDs) as a function of magnetic field for sequentially applied gate voltages, $V_g = 0$ V (initial state) (**c**), $V_g = +4$ V (**d**), $V_g = +5$ V (**e**), $V_g = -2$ V (**f**), and $V_g = -3$ V (**g**). $I_{dc} = 2.9$ mA. **h** Auto-oscillation spectra for $B = 0.56$ T with different gate voltages, extracted from Fig. 3c–g.

However, the maximum frequency tuning by current within this measurement range is ~0.26 GHz, which is much smaller than that by $V_g$ (~2.1 GHz). Furthermore, the slope of the oscillation frequency with respect to $I_{dc}$ is independent of the $V_g$, demonstrating that the current-induced SOT is not significantly affected by the $V_g$. This is confirmed by in-plane harmonic Hall measurements as well as the weakened non-linearity of our device (Supplementary Note 7). Note that the weak dependence of the slope of oscillation frequency versus the $I_{dc}$ curves on $V_g$ is different from the previous results[53,55,56]. This might be due to the different polar angles used for the measurements (Supplementary Note 8). We also note that the auto-oscillation dynamics in our samples belong to the propagation mode because the auto-oscillation frequency is higher than the FMR frequency[53,57,58] (Supplementary Note 9). Second, the $I_{th}$ is also modified by the $V_g$. Figure 4f shows the $I_{th}$ according to the $V_g$. Here, the $I_{th}$ values are extracted by a linear fit of the inverse of the PSD integral (Supplementary Note 10)[59]. The $I_{th}$ decreases (increases)

for positive (negative) $V_g$'s. Since the auto-oscillation occurs when SOT compensates the magnetic damping torque, the voltage-dependent $I_{th}$ is attributed to the voltage-controlled $\alpha_{eff}$[7,28,55,59], as shown in Fig. 2e. We remark that the leakage current due to the gate voltage is about a few pA (Supplementary Note 11), which is orders of magnitude smaller than the driving current of a few mA. We, therefore, conclude that its effect on the auto-oscillation frequency is negligible.

**Cumulative frequency control via a repetitive voltage pulse.** We finally demonstrate the emulation of synaptic plasticity[38–41], a key function in the learning process of neuromorphic devices, by exploiting the non-volatility of our voltage-controlled SHNO. For this experiment, we fabricated a 100-nm-width constriction SHNO device with a Ta (3 nm)/Pt(5 nm)/[Co(0.4 nm)/Ni(0.6 nm)]$_7$/Co(0.4 nm)/AlO$_x$(2 nm) structure. To verify cumulative frequency tuning, we applied a series of voltage pulses and measured the auto-oscillation spectrum between the pulses under

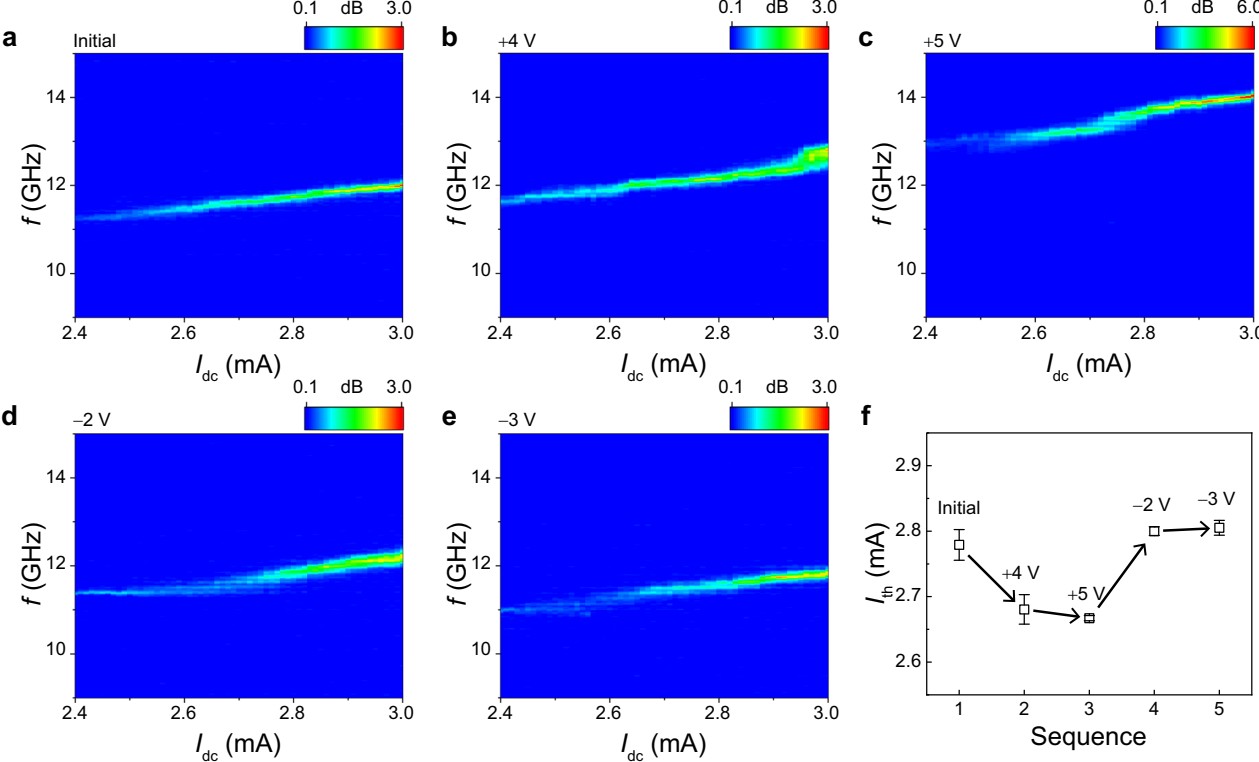

**Fig. 4 Current-dependent voltage-driven frequency tuning of SHNO. a–e** PSDs as a function of current for sequentially applied gate voltages, $V_g = 0\,V$ (initial state) (**a**), $V_g = +4\,V$ (**b**), $V_g = +5\,V$ (**c**), $V_g = -2\,V$ (**d**), and $V_g = -3\,V$ (**e**). $B = 0.56\,T$. **f**, Threshold current ($I_{th}$) according to the sequentially applied gate voltages, extracted from Fig. 4a–e. The error bars are due to the uncertainty in the linear fit of the data in Supplementary Fig. 13a–e.

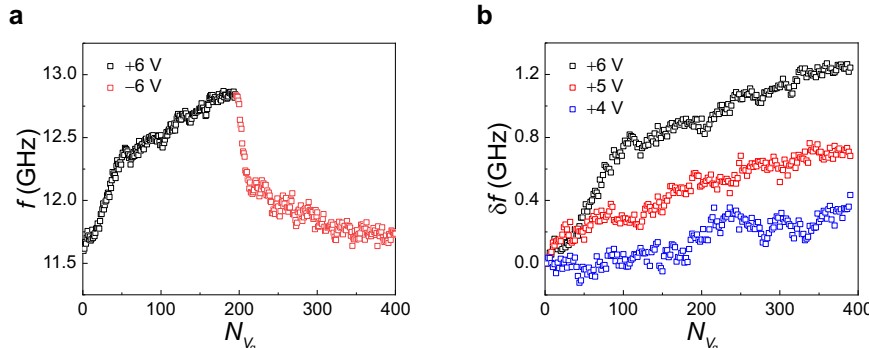

**Fig. 5 Cumulative frequency control via repetitive gate voltage pulses. a** The auto-oscillation frequency of the SHNO versus the number of positive (black) and negative (red) gate voltage pulses ($N_{V_g}$). **b** Voltage-driven cumulative frequency change ($\delta f$) versus $N_{V_g}$ for different gate voltage amplitudes.

a magnetic field ($B$) of 0.9 T applied in the direction with a polar angle $\theta$ of 80° and an azimuthal angle $\varphi$ of 70°. Here, a $V_g$ of 10 seconds was applied at room temperature while applying a current of 1.8 mA, which is different from the experiments shown in Figs. 2–4, where a $V_g$ was applied at 150 °C. Figure 5a shows the frequency change with the number of $V_g$ pulses ($N_{V_g}$) of ±6 V. The oscillation frequency gradually increases by ~1.24 GHz as the number of $V_g$ pulses of +6 V increases and is then restored to its initial value by subsequent negative $V_g$ pulses of −6 V. The frequency tuning for successive positive and negative voltage pulses mimics synaptic potentiation (strengthening in synaptic weight) and depression (weakening of synaptic weight), respectively. Figure 5b shows the cumulative frequency change ($\delta f$) for various $V_g$ amplitudes, demonstrating that the frequency change rate ($\delta f / N_{V_g}$) is effectively tuned by the magnitude of the $V_g$. The

voltage dependence of the frequency change rate can emulate the stimulus-dependent synaptic transition rate in a neuromorphic device. Furthermore, our device memorizes the modulated frequency with its non-volatile nature, offering a compact device layout that can store trained synaptic weights without the need for the separate memory circuitry required for conventional STOs[2,12].

## Discussion

We have demonstrated the voltage-driven GHz frequency tuning of SHNO with perpendicularly magnetized Pt/[Co/Ni]$_n$/Co/AlO$_x$ structures. It is found that the auto-oscillation frequency of the SHNO is effectively modulated up to 2.1 GHz by controlling the PMA with a $V_g$ of 5 V, which is equivalent to 1.25 MV/cm. Moreover, owing to the non-volatile nature of the gate effect, the cumulative oscillation frequency is controlled by repetitive

voltage pulses, which can mimic the biological synaptic functions of stimulus-dependent potentiation and depression. Therefore, our SHNO device can be utilized in the learning process of neuromorphic devices, thereby facilitating spin-based low-power neuromorphic hardware applications.

## Methods

**Sample preparation**. The thin films of Ta/Pt/[Co/Ni]$_n$/Co/AlO$_x$ structures were fabricated on high resistivity Si substrates by magnetron sputtering under a base pressure of $4.0 \times 10^{-6}$ Pa at room temperature. The metallic layers were deposited with an Ar gas pressure of 0.4 Pa and a dc power of 30 W, and the AlO$_x$ layer was formed by plasma oxidation of an Al layer with an O$_2$ pressure of 4.0 Pa and a dc power of 30 W for 75 s. The ZrO$_2$ gate oxide (40 nm) was grown at 125 °C by plasma enhanced atomic layer deposition (PE-ALD) using a TEMAZ [Tetra-kis(ethylmethylamido)zirconium] precursor. The oxygen plasma for the PE-ALD was formed with a rf power of 60 W and an O$_2$ gas flow of 500 sccm.

**PSD measurement**. All PSD measurements were carried out at room temperature using a home-built probe station where the sample was placed on an angle controllable holder located between two poles of an electromagnet. We used a bias-T to inject a dc current and to detect microwave signals simultaneously. A dc current was applied to the sample using a current source (Keithley 2450) with a current compliance of 3 mA that prevents sample degradation. The microwave signal generated from the sample was amplified by a low-noise amplifier (gain of +45 dB) and detected by a spectrum analyzer (Keysight N5173B). The resolution bandwidth and video bandwidth were set to 2 MHz and 9 kHz, respectively. The measured spectra were averaged at least 3 times to increase the signal-to-noise ratio.

**ST-FMR measurement**. ST-FMR measurements were performed at room temperature using the same probe station used in PSD measurements. For the ST-FMR measurements, we applied a magnetic field in the direction slightly tilted from the $z$ axis to attain an FMR rectified voltage. A microwave signal (power of +14 dBm) was injected into the sample by a signal generator (Keysight N9000B) through the RF port of a bias-T, and a dc voltage generated from the sample was detected by a lock-in amplifier (SR830). The tuning frequency was set to 313 Hz.

## Data availability

The data that support the findings of this study are available from the corresponding author upon reasonable request.

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

## Acknowledgements
This work is supported by the Samsung Research Funding Center of Samsung Electronics under Project Numbers SRFC-MA1702-02 and SRFC-MA1802-01. K.-J.K and B.-G.P acknowledge support from KAIST-funded Global Singularity Research Program for 2021 and the National Research Foundation of Korea (NRF) funded by the Korean Government (MSIP) [grant numbers: 2022M3I7A2079267, 2020R1A2C4001789, 2016R1A5A1008184].

## Author contributions
B.-G.P. and K.-J.K. planned and supervised the study. J.-G.C. and J.P. fabricated the devices and performed the experiment. M.-G.K. helps fabrication of the sample with a gate structure. D.K. and J.-S.R. help with the high-frequency measurement. J.-G.C., J.P., K.-J.K., K-J.L., and B.-G.P. analyzed the data and wrote the manuscript.

## Competing interests
The authors declare no competing interests.
