## [Peer Review File · Nature Communications]

Reviewers' Comments:

Reviewer #1:

Remarks to the Author:

In this manuscript, Choi et al. find a giant voltage-driven frequency tuning phenomenon and non-volatile feature in SHNO based on Ta/Pt/[Co/Ni]_n/Co/AlO_x structure. The value is much larger than the previously reported in-plane magnetized Pt/Py- and PMA W/CFB/MgO-based SHNOs. The reported voltage-control of SHNO dynamics in this work is clear and has significant improvement, which may facilitate the development of energy-efficient spin-based neuromorphic devices.

After reviewing the main text and its SM, I do not think this manuscript was carefully prepared, and the experimental results and behind mechanism were analyzed and discussed appropriately. To meet the criteria to be considered for publication in Nature Commun., the following points need to be addressed:

Major comments:

1. Although the sizeable non-volatile voltage control of the oscillation frequency of nano-magnet was achieved in nano-constriction SHNOs, and the experiment is not easy to successfully conducted, the experimental results and behind mechanism were not analyzed and discussed appropriately. The authors simply ascribed the significant frequency tuning to the general voltage-controlled magnetic anisotropy (VCMA) effect. As we well know, the nature of VCM includes the conventional mechanisms of charge, strain, and exchange coupling at the interfaces of heterostructures, as well as the emergent models of orbital reconstruction and electrochemical effect. Based on many previous reports about VCM (for example, ref-31 and 32) and the observed non-volatile and asymmetric VCM about positive and negative gating voltage (as well as elevated temperatures (150 oC) and long duration time (5 minutes) are required) in this work, the observed giant VCMA, similar to the previously reported FM/oxides systems, is likely relevant to the electric-field-driven O²⁻ displacement to modulate magnetism of the FM thin films by controlling interfacial or bulk oxidation states. Therefore, the authors should make a substantial analysis and discusses to point out the nature of the VCM in this work.

2. There is a significant change in the effective magnetic damping constant on the gating voltage. What is the reason that should be discussed?

3. The authors used the out-of-plane magnetic field geometry for the VCMA studies by using anomalous Hall resistance and ST-FMR, while adopted the near in-plane magnetic field orientation for the voltage-driven frequency tuning of SHNO. One can expect that the film may have different magnetic states (or different magnetic domain states) for in-plane and out-of-plane magnetic fields. What is the reason for selecting different field geometries? In addition, current-driven dynamics in SHNOs based on the strong PMA Pt/[Co/Ni]_n multilayer have been reported in the previous works, [e.g., PRL 114, 137201 (2015), PHYSICAL REVIEW APPLIED 11, 064038 (2019)]. The authors have to give suitable citations to the previous relevant works. In these previously reported works, three distinct current-induced dynamic modes were observed in the in-plane magnetic field orientation. Therefore, it should be helpful to the readers if the authors can provide some discussions about the different dynamics between this and the previous works.

4. During the VCMA experiments, there may have been some affections caused by the possible leak current and/or Joule heating effects. To eliminate these potential interferences, authors should provide the I-V curves between the ground electrode and the gating electrode

5. Based on the previous SHNOs experiments and theories of current-driven oscillation, it may be helpful to discuss which type of spin-wave was excited in this works based on the dispersion curve of the f vs. H and the current dependent oscillation frequency of SHNO.

6. There is a fatal flaw in developing energy-efficient spin-based neuromorphic devices by using the observed giant non-volatile voltage-driven frequency tuning of SHNO, which was highlighted and pointed out to the readers by the authors in this manuscript. Although voltage-driven frequency tuning is up to 2.1 GHz in their work, which is more than 40 times larger than the

previously reported values, the previous gating voltage control of SHNOs was the purely electrostatic response and had much faster operation speed and energy-efficient than this work. In this work, the observed more significant gating voltage effect was achieved at the elevated temperature (~ 150 oC) and long duration time (5 minutes), or the duration time ~ 10 seconds for voltage-induced cumulative frequency tuning due to the electric-field-driven oxygen ion displacement in the thin film. The non-volatile voltage control of the frequency of SHNO proved by this work is not a more energy-efficient approach than the previous reports, and even the current-based frequency tuning if the operation time (or duration time) of the gating voltage-induced accumulation effect cannot be reduced dramatically.

7. There are also miss the Celsius unit in several places in the main text.

Reviewer #2:

Remarks to the Author:

Despite minor suggestions written below, the manuscript "Voltage-driven gigahertz frequency tuning of spin Hall nano-oscillators" by Jong-Guk Choi, Jaehyeon Park, Min-Gu Kang, Doyoon Kim, Jae-Sung Rieh, Kyung-Jin Lee, Kab-Jin Kim and Byong-Guk Park should be accepted for publication as is, in its present form.

The manuscript "Voltage-driven gigahertz frequency tuning of spin Hall nano-oscillators" is devoted to the experimental study of the frequency tuning in a nano-constriction spin Hall nano-oscillator (SHNO) due to the voltage-controlled magnetic anisotropy effect. The results obtained by the authors evidently show that DC gate voltage (from -5 V to $+5$ V) applied to the nano-oscillator can substantially change its operating frequency. Moreover, the authors show that the action of repetitive gate voltage pulses can lead to the same effect of frequency tuning. The authors propose using the discovered effect to control the state of an individual SHNO in energy-efficient spin-based devices for neuromorphic computing, which makes the obtained results to be relevant for the electronics of the future.

The paper is technically sound and clearly written. All the main claims of the manuscript are understandably formulated and rather clearly explained. Important additional data are presented in the supplementary to the article.

The paper conclusions are based on evident experimental results. These results are explained by a simple but rather good theoretical description.

The obtained results are novel; they pave a new way for the implementation of systems for spin-based neuromorphic computing.

In general, the paper complies with the standards used by the scientific community and is important for the field of applied physics and spintronics.

I would like to give the authors only two quite minor recommendations, which in my opinion could improve the text:

1. Recently several review papers discussing neuromorphic computing have been published, namely:

Sangwan, V.K., Hersam, M.C. Neuromorphic nanoelectronic materials. *Nat. Nanotechnol.* (2020). <https://doi.org/10.1038/s41565-020-0647-z>

Grollier, J., Querlioz, D., Camsari, K.Y. et al. Neuromorphic spintronics. *Nat. Electron.* (2020). <https://doi.org/10.1038/s41928-019-0360-9>

Also, it seems to me it would be good to state in the paper that the spin-based neuromorphic devices can be made of antiferromagnets as well or at least cite one or two well-known recent papers in this field:

Khymyn, R., Lisenkov, I., Voorheis, J. et al. Ultra-fast artificial neuron: generation of picosecond-duration spikes in a current-driven antiferromagnetic auto-oscillator. *Sci. Rep.* 8, 15727 (2018) <https://doi.org/10.1038/s41598-018-33697-0>

Olga Sulymenko, Oleksandr Prokopenko, Ivan Lisenkov et al. Ultra-fast logic devices using artificial "neurons" based on antiferromagnetic pulse generators. *J. Appl. Phys.* 124, 152115 (2018)

<https://doi.org/10.1063/1.5042348>

I recommend the authors cite these papers in the manuscript.

2. In the PDF version of the paper symbol describing the units of temperature is shown incorrectly (see p. 5, 3 lines from the bottom of the page; p. 12, line 228).

Despite made remarks, the reviewed paper has a very good scientific quality. Its publication in Nature Communications is advisable.

Reviewer #3:

Remarks to the Author:

The manuscript titled "Voltage-driven gigahertz frequency tuning of spin Hall nano-oscillators", demonstrates a large non-volatile voltage-controlled frequency tuning in nano-constriction based spin Hall nano-oscillators (SHNOs). This work follows up the recently published work by Fulara et al in Nature Communications 11, 4006 (2020). In particular, authors employ different material combination to obtain a large frequency tuning of 2.1 GHz by an electric field of 1.25 MV/cm in SHNOs. The observed large frequency tuning is attributed to the well-known voltage-controlled magnetic anisotropy (VCMA) effect in perpendicularly magnetized Ta/Pt[Co/Ni]_n/Co/AlO_x structure. While I am not sure about the applicability of the material choice for practical applications, the paper is an important experimental advance and the obtained results are impressive for oscillator based neuromorphic computing. However, many important issues remain unclear and unanswered. Some of them are as follows:

1. If I understood correctly, in each ST-FMR/auto-oscillation measurement, authors applied gate voltage to the top electrode for 5 min at 150 C and subsequently carried out measurements after turning off the gate voltage. It is not clear why 150 C temp was needed to apply gate voltage and I wonder how the new voltage-driven state will decay over time after turning off the gate voltage? Should one also assume that there is no large leakage current involved during the application of voltage, since it is important for estimating the power consumption?
2. Authors have performed ST-FMR measurement on a bar shaped device under a perpendicular magnetic field and observed a large decrease in ST-FMR frequency and damping after applying +5V voltage. However, in auto-oscillation measurements, they observed somewhat opposite trend, which is increase in frequency under the application of +5V. Can the authors explain this opposite behavior?
3. Authors did not explain which type of VCMA effect is responsible for the observed effects? Namely, there are in general two types of VCMA effects. One is of electronic origin and the other mechanism is of ionic origin. Authors may like to benefit from the study available on arXiv: "Memristive control of mutual SHNO synchronization for neuromorphic computing", arXiv:2009.06594
4. It is also not clear why the authors specifically performed auto-oscillation measurements in close to in-plane magnetic fields, while they carried out ST-FMR measurements under perpendicular magnetic fields. The question arises because authors already observed a large frequency tuning and damping modulation in their ST-FMR measurement.
5. Authors did not explain the origin of large voltage-induced modulation of damping in their bar-shaped devices and therefore threshold current in SHNOs. Also, it remains unclear why authors chose a specific polar angle and an azimuthal angle in auto-oscillation measurements.
6. Looking at Fig.3(c-h), auto-oscillating signal dies off at higher fields for initial state and then starts to appear after applying positive gate voltage. However, there is no auto-oscillations at lower magnetic fields when the gate voltage is -3V. Is it because of the irreversible changes in the gated region after applying positive gate voltage? If yes, can the original state be restored in these devices?

7. In Fig. 4(a-e), slope of the frequency vs current is independent of gate voltage. Can the authors explain why the nonlinearity remains unaffected despite large changes in PMA? Authors may like to refer the following papers to address this.

Science Advances 5 (9), eaax8467 [DOI: 10.1126/sciadv.aax8467]
PHYSICAL REVIEW APPLIED 9, 014017 (2018) [DOI:
<https://doi.org/10.1103/PhysRevApplied.9.014017>]

8. There are few typos in the manuscript and I recommend authors to carefully proof-read the manuscript to improve the readability of the paper for a general audience. For ex- in page no 5 and 12, the unit after 150 seems missing.

Dear Reviewers,

We appreciate your comments and valuable queries, which have helped us improve the clarity and quality of our manuscript. Given below are detailed point-by-point responses to your questions and suggestions. The corresponding modifications are incorporated in the revised manuscript (marked in blue). We would like to ask your understanding of the delay in responding to the reviewer's comments due to additional experiments. We believe our manuscript has been improved significantly and now deserves publication in *Nature Communications*.

Yours sincerely,

Byong-Guk Park on behalf of all co-authors

Reviewer #1 (Remarks to the Author)

In this manuscript, Choi et al. find a giant voltage-driven frequency tuning phenomenon and non-volatile feature in SHNO based on Ta/Pt/[Co/Ni]_n/Co/AlO_x structure. The value is much larger than the previously reported in-plane magnetized Pt/Py- and PMA W/CFB/MgO-based SHNOs. The reported voltage-control of SHNO dynamics in this work is clear and has significant improvement, which may facilitate the development of energy-efficient spin-based neuromorphic devices. After reviewing the main text and its SM, I do not think this manuscript was carefully prepared, and the experimental results and behind mechanism were analyzed and discussed appropriately. To meet the criteria to be considered for publication in Nature Commun., the following points need to be addressed:

Response) We appreciate the reviewer's comment of "*The reported voltage-control of SHNO dynamics in this work is clear and has significant improvement, which may facilitate the development of energy-efficient spin-based neuromorphic devices*". However, she/he raised concerns about the analysis of the experimental results and the underlying mechanism. We below respond to the reviewer's comments with additional experimental and theoretical studies, which hopefully convince her/him that our work meets the criteria of *Nature Communications*.

1. Although the sizeable non-volatile voltage control of the oscillation frequency of nano-magnet was achieved in nano-constriction SHNOs, and the experiment is not easy to successfully conducted, the experimental results and behind mechanism were not analyzed and discussed appropriately. The authors simply ascribed the significant frequency tuning to the general voltage-controlled magnetic anisotropy (VCMA) effect. As we well know, the nature of VCM includes the conventional mechanisms of charge, strain, and exchange coupling at the interfaces of heterostructures, as well as the emergent models of orbital reconstruction and electrochemical effect. Based on many previous reports about VCM (for example, ref-31 and 32) and the observed non-volatile and asymmetric VCM about positive and negative gating voltage (as well as elevated temperatures (150 °C) and long duration time (5 minutes) are required) in this work, the observed giant VCMA, similar to the previously reported FM/oxides systems, is likely relevant to the electric-field-driven O²⁻ displacement to modulate magnetism of the FM thin films by controlling interfacial or bulk oxidation states. Therefore, the authors should make a substantial analysis and discusses to point out the nature of the VCM in this work.

Response) We thank the reviewer for this critical comment about the nature of voltage-controlled magnetic anisotropy (VCMA) effect in our samples. We agree with the reviewer that the VCMA effect in our sample is likely due to the electric-field-driven O^{2-} displacement. To verify the governing mechanism of the VCMA effect, we performed the following additional experiments: (i) measurement of resistance variation when a gate voltage is applied, (ii) comparison of the anisotropy changes by gate voltage application and oxidation time. As demonstrated below, these experimental results, along with the non-volatile nature, corroborate that the VCMA effect in our work is primarily due to the voltage-driven O^{2-} ion migration.

(i) Resistance variation by gate voltage

To verify the mechanism of the VCMA effect, we measured the resistance change of the SHNO device during the cumulative voltage-driven frequency tuning experiment. It is expected that the voltage-driven O^{2-} ion migration induces resistance changes through oxidation or reduction at the interface, unlike other mechanisms. To confirm this, we repeated the same measurement shown in Fig. 5 of the main text with a dc current (I_{dc}) of 2.1 mA and a voltage (V_g) duration time of 30 s. The top and bottom panels of Fig. R1(a) show the auto-oscillation frequency change with the number of V_g pulses (N_{V_g}) of +7 V and -5 V, respectively. The oscillation frequency gradually increases (decreases) by about 1.5 GHz with increasing N_{V_g} of +7 V (-5V), which is consistent with the result shown in Fig. 5(a) of the main text. While measuring the oscillation frequency with V_g , we monitored the resistance change $\frac{\Delta R}{R(0)}$ of the SHNO device. Here, $\frac{\Delta R}{R(0)} (\%) = \frac{R(N_{V_g}) - R(0)}{R(0)} \times 100$, where $R(0)$ is the initial resistance and $R(N_{V_g})$ is the resistance after applying N_{V_g} . Figure R1(b) shows $\frac{\Delta R}{R(0)}$ versus N_{V_g} for V_g 's of +7 V and -5 V. When the negative (positive) V_g is applied, the ΔR increases (decreases), which suggests that the negative (positive) voltage drives the O^{2-} ion toward (away from) the ferromagnet/oxide interface, causing oxidation (reduction) of the ferromagnet at the interface. We remark that the sample resistance is fully restored to the initial value when the opposite voltage is applied, which indicates that the resistance variation is not caused by sample degradation, for example by Joule heating.

Figure R1 (a) Top and bottom panels show the color plots of power spectral density as a function of N_{V_g} of +7 V and -5 V, respectively. (b) Resistance change ($\Delta R/R(0)$) versus N_{V_g} of +7 V and -5 V.

(ii) Comparison of the anisotropy changes by gate voltage application and by oxidation time

To further verify the VCMA mechanism, we measured the anisotropy variation with plasma oxidation time. Note that the negative voltage enhances the PMA of our Co/Ni film (Fig. 1(c) of the main text). To investigate whether the enhancement of the PMA is due to oxidation of ferromagnet at the interface, we prepared two Co/Ni samples with the same structure, but different O_2 plasma oxidation times ($t = 25s, 125s$). Figure R2(a) shows the hysteresis curves of the two Co/Ni samples with different oxidation states the ferromagnet/oxide interface. The sample with longer oxidation has a larger coercivity field than the sample with shorter oxidation, suggesting that the PMA increases with oxidation time.

The variation of PMA with oxidation time is further evaluated by ST-FMR measurements. Figure R2(b) shows the FMR frequency (f) as a function of resonance field B_{res} for the samples, from which we extracted the anisotropy field, B_k of each sample. Figure R2(c)

demonstrates the results that the sample with longer oxidation shows a larger B_k than the sample with shorter oxidation. This shows that the longer (shorter) oxidation has the same effect on the PMA as the negative (positive) voltage, corroborating that the VCMA effect in our sample is primarily dominated by the voltage-driven O^{2-} ion migration.

We have revised the manuscript as follows, and have added the above discussion in Supplementary Note 4.

On page 7, “*Note that the sample resistance increases (decreases) at negative (positive) voltages, suggesting that oxidation or reduction occurs at the ferromagnet/oxide interface when voltage is applied. In addition, a sample whose oxidation level is intentionally reduced (increased) by controlling the oxidation time has a low (high) PMA, which is the same effect as applying a positive (negative) gate voltage (Supplementary Note 4). This indicates that the VCMA effect in our samples is dominated by the voltage-driven oxygen ion migration.*”

Figure R2 (a) Magnetization hysteresis loops of two Co/Ni samples with out-of-plane magnetic field. Here, two samples have different plasma oxidation time, $t = 25$ s (black) and $t = 125$ s (red). (b) FMR frequency as a function of resonance field (B_{res}) for the two samples. The solid lines are the best fit based on the Kittel formula, $f_{res,z} = \frac{\gamma}{2\pi} (B_{res} + B_k)$, where γ is the gyromagnetic ratio. (c) Extracted anisotropy field B_k according to the oxidation time.

2. *There is a significant change in the effective magnetic damping constant on the gating voltage. What is the reason that should be discussed?*

Response) We thank the reviewer for this valuable comment. We believe that the change in effective magnetic damping by gate voltage is related to the voltage-induced modification of the interfacial oxidation state. As explained in the response to the question #1, the gate voltage modifies the interfacial oxidation states by the voltage-induced O^{2-} ion migration.

This would change the effective magnetic damping possibly through the modulation of interfacial Rashba spin-orbit-coupling (RSOC) as the surface (or interface) oxidation increases the RSOC [Phys. Rev. B **71**, 201403(R) (2005)] and the RSOC modifies the damping [Phys. Rev. Lett. **108**, 217202 (2012); Phys. Rev. B **87**, 054403 (2013)].

We further check whether interfacial oxidation alters the effective magnetic damping in our sample. Figure R3(a) shows the linewidth of the ST-FMR spectra for the samples with different oxidation times, which was obtained from the same measurements shown in Fig. R2. As summarized in Fig. R3(b), the effective damping constant α_{eff} of the sample with longer oxidation is larger than that of the sample with shorter oxidation. This demonstrates that the enhancement in effective magnetic damping is related to the interface oxidation, possibly due to the modulation of interfacial RSOC.

We have revised the manuscript as follows, and have added the above discussion in Supplementary Note 4.

On page 8, “The change in α_{eff} by gate voltage is also attributed to the voltage-induced modification of the interfacial oxidation state (Supplementary Note 4), possibly through the modulation of interfacial Rashba spin-orbit-coupling (RSOC) as the surface (or interface) oxidation increases the RSOC [49] and the RSOC modifies the damping [50,51].”

Figure R3 (a) Line width of ST-FMR spectrum as a function of driving frequency (f) for two Co/Ni samples with different plasma oxidation times, $t = 25$ s (black) and $t = 125$ s (red). Solid lines are the best fit based on Kittel formula, $\frac{d\Delta B}{df} = \frac{2\pi\alpha_{\text{eff}}}{\gamma}$, where the γ is the gyromagnetic ratio. (b) Extracted α_{eff} according to oxidation time.

3. The authors used the out-of-plane magnetic field geometry for the VCMA studies by using

anomalous Hall resistance and ST-FMR, while adopted the near in-plane magnetic field orientation for the voltage-driven frequency tuning of SHNO. One can expect that the film may have different magnetic states (or different magnetic domain states) for in-plane and out-of-plane magnetic fields. What is the reason for selecting different field geometries?

Response) We thank the reviewer for the valuable comment, which we overlooked in the original manuscript. The reason why we used different measurement geometries for the ST-FMR and SHNO experiments is that the purpose of each experiment is different. First, the ST-FMR measurement was performed to determine the PMA of the sample. As shown in previous literature [Phys. Rev. B **80**, 180415 (2009)], a precise measurement of PMA of Co/Ni films was achieved by performing FMR with an out-of-plane magnetic field, because it was not easy to accurately determine the PMA with an in-plane magnetic field due to the artifacts, such as local anisotropy variation or two-magnon scattering. On the other hand, we used an in-plane field geometry for the SHNO measurement because it is suitable for spin-orbit-torque-induced magnetization auto-oscillations. In this geometry, spin currents due to the spin Hall effect carry the in-plane spin polarization, which leads to magnetization oscillation when the spin-orbit torque (SOT) compensates for the in-plane magnetic-field induced damping torque. Therefore, the equilibrium magnetization should be in the plane. We note that the magnetic field was applied with a slight out-of-plane tilt to avoid SOT-driven magnetization switching [Appl. Phys. Lett. **111**, 032405 (2017)]

We note that the two experiments with different geometries are key to demonstrating our main idea of tuning the auto-oscillation frequency by modulating PMA with a gate voltage (Eq. (1) of the main text). Therefore, those measurement geometries are in line with our purpose.

Regarding the measurement geometry, we have added the following sentence in our revised manuscript.

On page 6, “*Note that the out-of-plane magnetic field geometry was employed to accurately determine the PMA, avoiding artifacts that might arise in the in-plane magnetic field geometry, such as local anisotropy variation or two magnon scattering [45].*”

On page 8, “*Here, we applied a magnetic field along the nearly in-plane direction to maximize the SOT-induced anti-damping effect. The slight tilt from the plane suppresses the SOT-induced magnetization switching [52]. The azimuthal angle was chosen to achieve a*

large electrical auto-oscillation signal [17,18].”

*In addition, current-driven dynamics in SHNOs based on the strong PMA Pt/[Co/Ni]*n* multilayer have been reported in the previous works, [e.g., PRL 114, 137201 (2015), PHYSICAL REVIEW APPLIED 11, 064038 (2019)]. The authors have to give suitable citations to the previous relevant works. In these previously reported works, three distinct current-induced dynamic modes were observed in the in-plane magnetic field orientation. Therefore, it should be helpful to the readers if the authors can provide some discussions about the different dynamics between this and the previous works.*

Response) We thank the reviewer for letting us know the important literatures. We have added the suggested papers as references of the revised manuscript (ref. #53,54).

As the reviewer pointed out, there are three different modes in SHNO; propagating spin wave mode, localized ‘bullet’ mode, and quasi-propagating mode. According to the literatures that the reviewer suggested, the propagating spin wave mode appears at a low current regime, and as the current increases, the localized ‘bullet’ mode starts to appear in an intermediate current regime and finally, the quasi-propagating mode that is weakly localized due to Oersted field is dominant at a high current regime. These three modes can be distinguished by comparing the auto-oscillation frequency, f_{AO} , with the ferromagnetic resonance frequency, f_{FMR} ; the (quasi) propagating spin wave mode is realized for $f_{AO} > f_{FMR}$ while the localized mode appears for $f_{AO} < f_{FMR}$. This is because the spatially localized mode with a frequency well below f_{FMR} has no well-defined real wave vector, \mathbf{k} , while the propagating mode with frequency higher than f_{FMR} can have a finite real \mathbf{k} . We therefore compared the obtained f_{AO} with f_{FMR} , and found that $f_{AO} < f_{FMR}$ for all current and gate voltage ranges explored in this work (the details of the comparison will be discussed in the response to the question #5 later). This suggests that the localized mode dominates the auto-oscillation in our SHNO device.

Note that this is different from the previous works that the reviewer suggested [Phys. Rev. Lett. **114**, 137201 (2015); Phys. Rev. Appl **11**, 064038 (2019)], which is possibly due to the different sample geometry. In the previous works, they used nano-gap structures, where there is no magnetic barrier isolating the active region from the surrounding film in the extended structure, so the coupling of the locally oscillating magnetization to spin waves is strong. Thus, the Slonczewski’s propagation mode with finite wave vector can be easily excited [J.

Magn. Magn. Mater. **159**, L1 (1996)]. On the other hand, our sample has a nano-constriction structure, where the magnetic film itself was patterned into a nano-sized constriction. In this structure, the auto-oscillation generally arises from the localized linear modes at the edge of the nano-constriction, especially for in-plane magnetic fields [Phys. Rev. Appl. **9**, 014017 (2018)]. We note that the propagating spin wave mode was recently reported even in a nano-constriction structure [Sci. Adv. **5**, eaax8467 (2019); Nat. Phys. **13**, 292 (2017)], but only when a magnetic field is applied along the nearly out-of-plane direction, which is different from our measurement geometry.

We have revised the manuscript as follows, and have added the above discussion in Supplementary Note 8.

On page 10, “*We also note that the auto-oscillation dynamics in our samples belongs to the non-propagating localized mode because the oscillation frequency lies well below the FMR frequency (Supplementary Note 8) [53,57,58].*”

4. During the VCMA experiments, there may have been some affections caused by the possible leak current and/or Joule heating effects. To eliminate these potential interferences, authors should provide the I-V curves between the ground electrode and the gating electrode

Response) We appreciate the reviewer’s comment on the leak current and associated Joule heating effects. We measured the leak current (I_{leak}) as a function of gate voltage (V_g) in the Ta (3 nm)/Pt (5 nm)/[Co (0.45 nm)/Ni (0.6 nm)]₇/Co (0.45 nm)/AlO_x (2 nm)/Ta(3 nm)/ZrO₂(40 nm) structure, which is the same sample shown in Fig. 1 of the main text. Figure R4 shows the I - V curve, demonstrating that I_{leak} is less than 3 pA within our experimental V_g range of -5 V to +5 V. Since the I_{leak} is 10^9 times smaller than the threshold current for auto-oscillation of a few mA, we believe that the leak current does not significantly affect the device properties nor induce Joule heating. In addition, in our experiments, we always measured the auto-oscillation spectra with the gate voltage turned off, thus the leak current does not affect the auto-oscillation measurement.

We have revised the manuscript as follows, and have added the above discussion in Supplementary Note 10.

On page 10, “*We remark that the leakage current due to the gate voltage is about a few pA [Supplementary Note 10], which is orders of magnitude smaller than the driving current of a*

few mA. We therefore conclude that its effect on the auto-oscillation frequency is negligible.”

Figure R4 The leakage current I_{leak} due to the gate voltage V_g for the Ta (3 nm)/Pt (5 nm)/[Co (0.45 nm)/Ni (0.6 nm)]₇/Co (0.45 nm)/AlO_x (2 nm)/Ta(3 nm)/ZrO₂(40 nm) structure.

5. Based on the previous SHNOs experiments and theories of current-driven oscillation, it may be helpful to discuss which type of spin-wave was excited in this work based on the dispersion curve of the f vs. H and the current dependent oscillation frequency of SHNO.

Response) We thank the reviewer for the valuable comment. As we explained in the response to the question #3, the mode of auto-oscillation can be distinguished by comparing the auto-oscillation frequency (f_{AO}) and ferromagnetic resonance frequency (f_{FMR}); the propagation spin wave mode (localized mode) is dominant for $f_{\text{AO}} > f_{\text{FMR}}$ ($f_{\text{AO}} < f_{\text{FMR}}$). Here we provide the detailed comparison between f_{AO} and f_{FMR} .

The f_{FMR} can be calculated using the formula, $f_{\text{FMR}} = \frac{\gamma}{2\pi} \sqrt{B_{\text{int}}(B_{\text{int}} - B_k \sin \theta_{\text{int}})}$, where B_k is the perpendicular magnetic anisotropy field, which is extracted from Fig. 2 of the main text; B_{int} and θ_{int} are the effective magnetic field and polar angle of the magnetization direction, respectively, which can be determined by the equations $B \sin \theta = B_{\text{int}} \sin \theta_{\text{int}}$ and $B \cos \theta = (B_{\text{int}} - B_k) \cos \theta_{\text{int}}$ [Phys. Rev. B **97**, 184402 (2018); Sci. Adv. **5**, eaax8467 (2019)]. Here, B and θ are the magnitude and polar angle of the external magnetic field used for the auto-oscillation measurements, respectively, $B = 0.56$ T and $\theta = 80^\circ$ in our measurement geometry. Figure R5 shows the calculated f_{FMR} for various B_k ranging from -0.08 T to 0.16 T, which can be obtained in our sample depending on gate voltage. The f_{FMR} increases with decreasing the B_k , which is consistent with the schematic

shown in Fig. 1b of the main text. It is clearly demonstrated that all f_{FMR} values are much larger than the measured f_{AO} depicted by the closed symbols in Fig. R5. We, therefore, conclude that the spin wave in our SHNO device is excited in localized mode.

We have revised the manuscript as follows, and have added the above discussion in Supplementary Note 8.

On page 10, “We note that the auto-oscillation dynamics in our samples belongs to the non-propagating localized mode because the oscillation frequency lies well below the FMR frequency [Supplementary Note 8] [53,57,58].”

Figure R5 (solid lines) Calculated ferromagnetic resonance frequency (f_{FMR}) for various perpendicular magnetic anisotropy fields (B_k) in the range between 0.16 T and -0.08 T with 0.02 T steps. (closed symbols) Measured auto-oscillation frequency ($f_{\text{A.O.}}$) as a function of driving current for several gate voltages, which was extracted from Figs. 4(a-e).

6. There is a fatal flaw in developing energy-efficient spin-based neuromorphic devices by using the observed giant non-volatile voltage-driven frequency tuning of SHNO, which was highlighted and pointed out to the readers by the authors in this manuscript. Although voltage-driven frequency tuning is up to 2.1 GHz in their work, which is more than 40 times larger than the previously reported values, the previous gating voltage control of SHNOs was the purely electrostatic response and had much faster operation speed and energy-efficient than this work. In this work, the observed more significant gating voltage effect was achieved

at the elevated temperature (~150 °C) and long duration time (5 minutes), or the duration time~ 10 seconds for voltage-induced cumulative frequency tuning due to the electric-field-driven oxygen ion displacement in the thin film. The non-volatile voltage control of the frequency of SHNO proved by this work is not a more energy-efficient approach than the previous reports, and even the current-based frequency tuning if the operation time (or duration time) of the gating voltage-induced accumulation effect cannot be reduced dramatically.

Response) We thank the reviewer for this critical comment on the viability of the VCMA effect in practical applications. In our study, we used a 40 nm thick gate oxide (ZrO_2) for which a gate voltage needs to be applied for a few seconds to minutes at 150°C to observe a sizable electric field effect. We agree with the reviewer that these conditions are not applicable in real devices. However, we below show that this is not a fundamental limit of our device but could be further improved by materials engineering. To verify this argument, we tested another sample of Ta(3 nm)/Pt(5 nm)/[Co(0.45 nm)/Ni(0.6 nm)]₃/Co(0.45 nm)/AlO_x(2 nm) structure with a thin gate oxide of ZrO_2 (5 nm). Figure R6 shows the anomalous Hall resistance (R_{xy}) as a function of perpendicular magnetic field (B_z), demonstrating that the coercivity of the sample is reduced by a gate voltage ($V_g = 20$ V) applied at room temperature for 1 ms. Furthermore, we showed in the previous report [Nat. Commun. **12**, 7111 (2021)] that the operation speed of the VCMA effect at room temperature decreased to 20 μs by introducing double gate oxide of TiO_2 (2 nm)/ ZrO_2 (5 nm) structure. These results indicate that the operation time in our sample can be further improved by material engineering. We note that the switching speed of an order of sub *ns* has been demonstrated in the ReRAM technology, whose switching mechanism is also the local movement of the oxygen ions as for our device [IEEE J. Solid-State Circuits **48**, 878-891 (2013); Adv. Electron. Mater. **3**, 1700263 (2017)]. Given that the device properties will be improved through further material developments, we believe that the proposed large frequency modulation by the VCMA effect has potential for applications as neuromorphic devices.

We have inserted following sentence in the conclusion part of our manuscript.

On page 6, “*We note that the VCMA effect can be improved by materials engineering, as it has been demonstrated that the gate effect occurs at room temperature and short voltage pulses by introducing a thin or double gate oxide (Supplementary Note 2) [44].*”

Figure R6. Anomalous Hall resistance (R_{xy}) as a function of perpendicular magnetic field (B_z) in a Ta (3 nm)/Pt (5 nm)/[Co (0.45 nm)/Ni (0.6 nm)]₃/Co (0.45 nm)/AlOx (2 nm)/ZrO₂(5 nm) sample when applying a V_g of +20 V for 1 ms at room temperature.

7. There are also miss the Celsius unit in several places in the main text.

Response) We apologize for missing the unit. We have inserted the Celsius unit on page 5 and page 11 of the revised manuscript.

Reviewer #2 (Remarks to the Author)

Despite minor suggestions written below, the manuscript “Voltage-driven gigahertz frequency tuning of spin Hall nano-oscillators” by Jong-Guk Choi, Jaehyeon Park, Min-Gu Kang, Doyoon Kim, Jae-Sung Rieh, Kyung-Jin Lee, Kab-Jin Kim and Byong-Guk Park should be accepted for publication as is, in its present form. The manuscript “Voltage-driven gigahertz frequency tuning of spin Hall nano-oscillators” is devoted to the experimental study of the frequency tuning in a nano-constriction spin Hall nano-oscillator (SHNO) due to the voltage-controlled magnetic anisotropy effect. The results obtained by the authors evidently show that DC gate voltage (from -5 V to $+5$ V) applied to the nano-oscillator can substantially change its operating frequency. Moreover, the authors show that the action of repetitive gate voltage pulses can lead to the same effect of frequency tuning. The authors propose using the discovered effect to control the state of an individual SHNO in energy-efficient spin-based devices for neuromorphic computing, which makes the obtained results to be relevant for the electronics of the future. The paper is technically sound and clearly written. All the main claims of the manuscript are understandably formulated and rather clearly explained. Important additional data are presented in the supplementary to the article. The paper conclusions are based on evident experimental results. These results are explained by a simple but rather good theoretical description. The obtained results are novel; they pave a new way for the implementation of systems for spin-based neuromorphic computing. In general, the paper complies with the standards used by the scientific community and is important for the field of applied physics and spintronics. I would like to give the authors only two quite minor recommendations, which in my opinion could improve the text:

Response) We appreciate the reviewer for recommending publication of our manuscript in the present form. Moreover, she/he acknowledged our study as “*The paper is technically sound and clearly written. All the main claims of the manuscript are understandably formulated and rather clearly explained. Important additional data are presented in the supplementary to the article*” and “*the paper complies with the standards used by the scientific community and is important for the field of applied physics and spintronics*”. We below respond to the reviewer’s additional comments, which have been reflected in the revised manuscript. This helped to further improve the manuscript.

1. Recently several review papers discussing neuromorphic computing have been published,

namely: Sangwan, V.K., Hersam, M.C. *Neuromorphic nanoelectronic materials*. *Nat. Nanotechnol.* (2020). <https://doi.org/10.1038/s41565-020-0647-z>

Grollier, J., Querlioz, D., Camsari, K.Y. et al. *Neuromorphic spintronics*. *Nat. Electron.* (2020). <https://doi.org/10.1038/s41928-019-0360-9>

Also, it seems to me it would be good to state in the paper that the spin-based neuromorphic devices can be made of antiferromagnets as well or at least cite one or two well-known recent papers in this field:

Khymyn, R., Lisenkov, I., Voorheis, J. et al. *Ultra-fast artificial neuron: generation of picosecond-duration spikes in a current-driven antiferromagnetic auto-oscillator*. *Sci. Rep.* 8, 15727 (2018) <https://doi.org/10.1038/s41598-018-33697-0>

Olga Sulymenko, Oleksandr Prokopenko, Ivan Lisenkov et al. *Ultra-fast logic devices using artificial “neurons” based on antiferromagnetic pulse generators*. *J. Appl. Phys.* 124, 152115 (2018) <https://doi.org/10.1063/1.5042348>

I recommend the authors cite these papers in the manuscript.

Response) We thank the reviewer for letting us know the important literatures. We have added the suggested papers as references of the revised manuscript. (refs. # 3-6)

2. *In the PDF version of the paper symbol describing the units of temperature is shown incorrectly (see p. 5, 3 lines from the bottom of the page; p. 12, line 228). Despite made remarks, the reviewed paper has a very good scientific quality. Its publication in Nature Communications is advisable.*

Response) We apologize for missing the unit. We have inserted the Celsius unit on page 5 and page 11 of the revised manuscript.

Reviewer #3 (Remarks to the Author)

The manuscript titled “Voltage-driven gigahertz frequency tuning of spin Hall nano-oscillators”, demonstrates a large non-volatile voltage-controlled frequency tuning in nano-constriction based spin Hall nano-oscillators (SHNOs). This work follows up the recently published work by Fulara et al in Nature Communications 11, 4006 (2020). In particular, authors employ different material combination to obtain a large frequency tuning of 2.1 GHz by an electric field of 1.25 MV/cm in SHNOs. The observed large frequency tuning is attributed to the well-known voltage-controlled magnetic anisotropy (VCMA) effect in perpendicularly magnetized Ta/Pt[Co/Ni]_n/Co/AlO_x structure. While I am not sure about the applicability of the material choice for practical applications, the paper is an important experimental advance and the obtained results are impressive for oscillator based neuromorphic computing. However, many important issues remain unclear and unanswered. Some of them are as follows:

Response) First of all, we appreciate the reviewer’s comment that “*the paper is an important experimental advance and the obtained results are impressive for oscillator based neuromorphic computing*”. We below respond to the reviewer’s additional comments below, which hopefully alleviates the reviewer’s concerns and the revised manuscript is acceptable for publication.

1. If I understood correctly, in each ST-FMR/auto-oscillation measurement, authors applied gate voltage to the top electrode for 5 min at 150 °C and subsequently carried out measurements after turning off the gate voltage. It is not clear why 150 °C temp was needed to apply gate voltage and I wonder how the new voltage-driven state will decay over time after turning off the gate voltage?

Response) We thank the reviewer for this fruitful comment. As will be discussed later in the response to the question #3, the non-volatile VCMA effect in our device originates from the electric-field-driven O²⁻ ion migration. As the ion migration occurs through the thermal activation process [Phys. Rev. Lett. **113**, 267202 (2014); Nat. Mater. **14** 174-181 (2015)], we applied a gate voltage at 150 °C to obtain sufficient VCMA effect. We note that these conditions are not applicable in real devices. However, we below show that this is not a fundamental limit of our device but could be further improved by materials engineering. To

verify this argument, we tested another sample of Ta(3 nm)/Pt(5 nm)/[Co(0.45 nm)/Ni(0.6 nm)]₃/Co(0.45 nm)/AlO_x(2 nm) structure with a thin gate oxide of ZrO₂ (5 nm). Figure R7 shows the anomalous Hall resistance (R_{xy}) as a function of perpendicular magnetic field (B_z), demonstrating that the coercivity of the sample is reduced by a gate voltage ($V_g = 20$ V) applied at room temperature for 1 ms. Furthermore, we showed in the previous report [Nat. Commun. **12**, 7111 (2021)] that the operation speed of the VCMA effect at room temperature decreased to 20 μ s by introducing double gate oxide of TiO₂(2 nm)/ZrO₂(5 nm) structure. These results indicate that the operation time in our sample can be further improved by material engineering. We note that the switching speed of an order of sub *ns* has been demonstrated in the ReRAM technology, whose switching mechanism is also the local movement of the oxygen ions as for our device [IEEE J. Solid-State Circuits **48**, 878-891 (2013); Adv. Electron. Mater. **3**, 1700263 (2017)]. Given that the device properties will be improved through further material developments, we believe that the proposed large frequency modulation by the VCMA effect has potential for applications as neuromorphic devices.

To demonstrate the retention of the voltage-driven state, we measured how long the voltage-controlled frequency lasts after turning off the gate voltage. Figure R8 shows auto-oscillation frequency as a function of time for our Co/Ni sample. Here, we used the same experimental conditions as Figs. 3 and 4 of the main text. The result shows that the frequency is almost constant up to 10⁴ seconds both for +5V (red) and for -5V (black). We note that this retention time is comparable to the characteristic time scale in previously reported memristive devices [Sci. Rep. **9**, 6144 (2019); Sci. Rep. **10**, 2807 (2020); Nat. Commun. **12**, 2968 (2021)]. This indicates that the voltage-driven frequency modulation can be utilized in long term plasticity of the neuromorphic device [Nature **361**, 31 (1993); Nature **431**, 796 (2004)].

We have added the above discussion in Supplementary Note 1 & 2 and mentioned it in the main text as follows.

On page 5, “We note that the voltage-driven state is maintained after turning off the V_g [32,33,42,43] (Supplementary Note 1).”

On page 6, “We note that the VCMA effect can be improved by materials engineering, as it has been demonstrated that the gate effect occurs at room temperature and short voltage pulses by introducing a thin or double gate oxide (Supplementary Note 2) [44].”

Figure R7. Anomalous Hall resistance (R_{xy}) as a function of perpendicular magnetic field (B_z) in a Ta (3 nm)/Pt (5 nm)/[Co (0.45 nm)/Ni (0.6 nm)]₃/Co (0.45 nm)/AlO_x (2 nm)/ZrO₂(5 nm) sample when applying a V_g of +20 V for 1 ms at room temperature.

Figure R8 Temporal variation of auto-oscillation frequency after turning off the gate voltage of +5 V (red) and -5 V (black) for the Ta (3 nm)/Pt (5 nm)/[Co (0.4 nm)/Ni (0.6 nm)]₇/Co (0.4 nm)/AlO_x (2 nm)/Ta(3 nm)/ZrO₂(40 nm) sample.

Should one also assume that there is no large leakage current involved during the application of voltage, since it is important for estimating the power consumption?

Response) We appreciate the reviewer's comment on the leak current and associated Joule heating effects. We measured the leak current (I_{leak}) as a function of gate voltage (V_g) in the Ta (3 nm)/Pt (5 nm)/[Co (0.45 nm)/Ni (0.6 nm)]₇/Co (0.45 nm)/AlO_x (2 nm)/Ta(3 nm)/ZrO₂(40 nm) structure, which is the same sample shown in Fig. 1 of the main text. Figure R9 shows the I - V curve, demonstrating that I_{leak} is less than 3 pA within our

experimental V_g range of -5 V to +5 V. Since the I_{leak} is 10^9 times smaller than the threshold current for auto-oscillation of a few mA, we believe that the leak current does not significantly affect the device properties nor induce Joule heating. In addition, in our experiments, we always measured the auto-oscillation spectra with the gate voltage turned off, thus the leak current does not affect the auto-oscillation measurement.

We have revised the manuscript as follows, and have added the above discussion in Supplementary Note 10.

On page 10, “We remark that the leakage current due to the gate voltage is about a few pA (Supplementary Note 10), which is orders of magnitude smaller than the driving current of a few mA. We therefore conclude that its effect on the auto-oscillation frequency is negligible.”

Figure R9 The leakage current I_{leak} due to the gate voltage V_g for the Ta (3 nm)/Pt (5 nm)/[Co (0.45 nm)/Ni (0.6 nm)]₇/Co (0.45 nm)/AlO_x (2 nm)/Ta(3 nm)/ZrO₂(40 nm) structure.

2. Authors have performed ST-FMR measurement on a bar shaped device under a perpendicular magnetic field and observed a large decrease in ST-FMR frequency and damping after applying +5V voltage. However, in auto-oscillation measurements, they observed somewhat opposite trend, which is increase in frequency under the application of +5V. Can the authors explain this opposite behavior?

Response) We thank the reviewer for the comment. The difference arises from the different measurement geometry. In our experiment, the ST-FMR was performed under an *out-of-plane magnetic field* to precisely determine perpendicular magnetic anisotropy (PMA), because it was not easy to accurately determine the PMA value with an in-plane magnetic

field due to the artifacts, such as local anisotropy variation or two-magnon scattering [Phys. Rev. B **80**, 180415 (2009)]. On the other hand, the auto-oscillation was measured under an *in-plane magnetic field* because it is suitable for spin-orbit-torque-induced magnetization auto-oscillations. In this geometry, spin currents due to the spin Hall effect carry the in-plane spin polarization, which leads to magnetization oscillation when the spin-orbit torque (SOT) compensates for the in-plane magnetic-field induced damping torque. Therefore, the equilibrium magnetization should be in the plane.

In the presence of an in-plane field B_{\parallel} , the resonance frequency (f_{res}) of a PMA material is determined as

$$f_{\text{res}} = \frac{\gamma}{2\pi} \sqrt{B_{\text{k}}^2 - B_{\parallel}^2} \quad (0 < B_{\parallel} < B_{\text{k}}),$$

$$f_{\text{res}} = \frac{\gamma}{2\pi} \sqrt{B_{\parallel}(B_{\parallel} - B_{\text{k}})} \quad (B_{\parallel} \geq B_{\text{k}}), \quad (\text{R1})$$

where γ is the gyromagnetic ratio, B_{k} is the effective perpendicular anisotropy field. Figure R10 shows the schematic plot of Eq. (R1). For $B_{\parallel} = 0$ at which magnetization is perpendicularly aligned (ST-FMR measurement), a higher PMA gives a larger resonance frequency. However, for large B_{\parallel} with magnetization close to in-plane (auto-oscillation experiment), the opposite trend is obtained because the PMA prevents the magnetization from aligning to the plane. This can explain the opposite behavior of the resonance frequency variations between the ST-FMR and auto-oscillation measurements.

Figure R10 The resonance frequency (f_{res}) as a function of in-plane magnetic field (B_{\parallel}) for high PMA (blue) and low PMA (red).

3. Authors did not explain which type of VCMA effect is responsible for the observed effects? Namely, there are in general two types of VCMA effects. One is of electronic origin and the other mechanism is of ionic origin. Authors may like to benefit from the study available on arXiv: “Memristive control of mutual SHNO synchronization for neuromorphic computing”, arXiv:2009.06594

Response) We thank the reviewer for this critical comment about the nature of voltage-controlled magnetic anisotropy (VCMA) effect in our samples. We agree with the reviewer that the VCMA effect in our sample is likely due to the electric-field-driven O^{2-} displacement. To verify the governing mechanism of the VCMA effect, we performed the following additional experiments: (i) measurement of resistance variation when a gate voltage is applied, (ii) comparison of the anisotropy changes by gate voltage application and oxidation time. As demonstrated below, these experimental results, along with the non-volatile nature, corroborate that the VCMA effect in our work is primarily due to the voltage-driven O^{2-} ion migration.

(i) Resistance variation by gate voltage

To verify the mechanism of the VCMA effect, we measured the resistance change of the SHNO device during the cumulative voltage-driven frequency tuning experiment. It is expected that the voltage-driven O^{2-} ion migration induces resistance changes through oxidation or reduction at the interface, unlike other mechanisms. To confirm this, we repeated the same measurement shown in Fig. 5 of the main text with a dc current (I_{dc}) of 2.1 mA and a voltage (V_g) duration time of 30 s. The top and bottom panels of Fig. R11(a) show the auto-oscillation frequency change with the number of V_g pulses (N_{V_g}) of +7 V and -5 V, respectively. The oscillation frequency gradually increases (decreases) by about 1.5 GHz with increasing N_{V_g} of +7 V (-5V), which is consistent with the result shown in Fig. 5(a) of the main text. While measuring the oscillation frequency with V_g , we monitored the resistance change $\frac{\Delta R}{R(0)}$ of the SHNO device. Here, $\frac{\Delta R}{R(0)} (\%) = \frac{R(N_{V_g}) - R(0)}{R(0)} \times 100$, where $R(0)$ is the initial resistance and $R(N_{V_g})$ is the resistance after applying N_{V_g} . Figure R11(b) shows $\frac{\Delta R}{R(0)}$ versus N_{V_g} for V_g 's of +7 V and -5 V. When the negative (positive) V_g is applied, the ΔR increases (decreases), which suggests that the negative (positive) voltage drives the O^{2-} ion toward (away from) the ferromagnet/oxide interface, causing oxidation (reduction) of the

ferromagnet at the interface. We remark that the sample resistance is fully restored to the initial value when the opposite voltage is applied, which indicates that the resistance variation is not caused by sample degradation, for example by Joule heating.

Figure R11 (a) Top and bottom panels show the color plots of power spectral density as a function of N_{V_g} of +7 V and -5 V, respectively. (b) Resistance change ($\Delta R/R(0)$) versus N_{V_g} of +7 V and -5 V.

(ii) Comparison of the anisotropy changes by gate voltage application and by oxidation time

To further verify the VCMA mechanism, we measured the anisotropy variation with plasma oxidation time. Note that the negative voltage enhances the PMA of our Co/Ni film (Fig. 1(c) of the main text). To investigate whether the enhancement of the PMA is due to oxidation of ferromagnet at the interface, we prepared two Co/Ni samples with the same structure, but different O_2 plasma oxidation times ($t = 25s, 125s$). Figure R12(a) shows the hysteresis curves of the two Co/Ni samples with different oxidation states the ferromagnet/oxide interface. The sample with longer oxidation has a larger coercivity field than the sample with shorter oxidation, suggesting that the PMA increases with oxidation

time.

The variation of PMA with oxidation time is further evaluated by ST-FMR measurements. Figure R12(b) shows the FMR frequency (f) as a function of resonance field B_{res} for the samples, from which we extracted the anisotropy field, B_k of each sample. Figure R12(c) demonstrates the results that the sample with longer oxidation shows a larger B_k than the sample with shorter oxidation. This shows that the longer (shorter) oxidation has the same effect on the PMA as the negative (positive) voltage, corroborating that the VCMA effect in our sample is primarily dominated by the voltage-driven O^{2-} ion migration.

We have revised the manuscript as follows, and have added the above discussion in Supplementary Note 4. We have also added the suggested papers [Nat. Mater. **21**, 81-87 (2021); publication version of *arXiv:2009.06594*] in reference of the revised manuscript (ref. #23).

On page 7, “Note that the sample resistance increases (decreases) at negative (positive) voltages, suggesting that oxidation or reduction occurs at the ferromagnet/oxide interface when voltage is applied. In addition, a sample whose oxidation level is intentionally reduced (increased) by controlling the oxidation time has a low (high) PMA, which is the same effect as applying a positive (negative) gate voltage (Supplementary Note 4). This indicates that the VCMA effect in our samples is dominated by the voltage-driven oxygen ion migration.”

Figure R12 (a) Magnetization hysteresis loops of two Co/Ni samples with out-of-plane magnetic field. Here, two samples have different plasma oxidation time, $t = 25$ s (black) and $t = 125$ s (red). (b) FMR frequency as a function of resonance field (B_{res}) for the two samples. The solid lines are the best fit based on the Kittel formula, $f_{res,z} = \frac{\gamma}{2\pi} (B_{res} + B_k)$, where γ is the gyromagnetic ratio. (c) Extracted anisotropy field B_k according to the oxidation time.

4. It is also not clear why the authors specifically performed auto-oscillation measurements

in close to in-plane magnetic fields, while they carried out ST-FMR measurements under perpendicular magnetic fields. The question arises because authors already observed a large frequency tuning and damping modulation in their ST-FMR measurement.

Response) We thank the reviewer for the fruitful comment, which we overlooked in the original manuscript. The reason why we used different measurement geometries for the ST-FMR and SHNO experiments is that the purpose of each experiment is different. First, the ST-FMR measurement was performed to determine the PMA of the sample. As shown in previous literature [Phys. Rev. B **80**, 180415 (2009)], a precise measurement of PMA of Co/Ni films was achieved by performing FMR with an out-of-plane magnetic field, because it was not easy to accurately determine the PMA with an in-plane magnetic field due to the artifacts, such as local anisotropy variation or two magnon scattering. On the other hand, we used an in-plane field geometry for the SHNO measurement because it is suitable for spin-orbit-torque-induced magnetization auto-oscillations. In this geometry, spin currents due to the spin hall effect carry the in-plane spin polarization, which leads to magnetization oscillation when the spin-orbit torque (SOT) compensates for the in-plane magnetic-field induced damping torque. Therefore, the equilibrium magnetization should be in the plane. We note that the magnetic field was applied with a slight out-of-plane tilt to avoid SOT-driven magnetization switching [Appl. Phys. Lett. **111**, 032405 (2017)]

We note that the two experiments with different geometries are key to demonstrating our main idea of tuning the auto-oscillation frequency by modulating PMA with a gate voltage (Eq. (1) of the main text). Therefore, those measurement geometries are in line with our purpose.

Regarding the measurement geometry, we have added the following sentence in our revised manuscript.

On page 6, “*Note that the out-of-plane magnetic field geometry was employed to accurately determine the PMA, avoiding artifacts that might arise in the in-plane magnetic field geometry, such as local anisotropy variation or two magnon scattering [45].*”

On page 8, “*Here, we applied a magnetic field along the nearly in-plane direction to maximize the SOT-induced anti-damping effect. The slight tilt from the plane suppresses the SOT-induced magnetization switching [52]. The azimuthal angle was chosen to achieve a large electrical auto-oscillation signal [17,18].*”

5. Authors did not explain the origin of large voltage-induced modulation of damping in their bar-shaped devices and therefore threshold current in SHNOs.

Response) We thank the reviewer for this valuable comment. We believe that the change in effective magnetic damping by gate voltage is related to the voltage-induced modification of the interfacial oxidation state. As explained in the response to the question #3, the gate voltage modifies the interfacial oxidation states by the voltage-induced O^{2-} ion migration. This would change the effective magnetic damping possibly through the modulation of interfacial Rashba spin-orbit-coupling (RSOC) as the surface (or interface) oxidation increases the RSOC [Phys. Rev. B **71**, 201403(R) (2005)] and the RSOC modifies the damping [Phys. Rev. Lett. **108**, 217202 (2012); Phys. Rev. B **87**, 054403 (2013)].

We further check whether interfacial oxidation alters the effective magnetic damping in our sample. Figure R13(a) shows the linewidth of the ST-FMR spectra for the samples with different oxidation times, which was obtained from the same measurements shown in Fig. R12. As summarized in Fig. R13(b), the effective damping constant α_{eff} of the sample with longer oxidation is larger than that of the sample with shorter oxidation. This demonstrates that the enhancement in effective magnetic damping is related to the interface oxidation, possibly due to the modulation of interfacial RSOC.

We have revised the manuscript as follows, and have added the above discussion in Supplementary Note 3.

On page 8, “The change in α_{eff} by gate voltage is also attributed to the voltage-induced modification of the interfacial oxidation state (Supplementary Note 3), possibly through the modulation of interfacial Rashba spin-orbit-coupling (SOC) as the surface (or interface) oxidation increases the RSOC [49] and the RSOC modifies the damping [50,51].”

Figure R13 (a) Line width of ST-FMR spectrum as a function of driving frequency (f) for two Co/Ni samples with different plasma oxidation times, $t = 25$ s (black) and $t = 125$ s (red). Solid lines are the best fit based on Kittel formula, $\frac{d\Delta B}{df} = \frac{2\pi\alpha_{\text{eff}}}{\gamma}$, where the γ is the gyromagnetic ratio. (b) Extracted α_{eff} according to oxidation time.

Also, it remains unclear why authors chose a specific polar angle and an azimuthal angle in auto-oscillation measurements.

Response) In our SHNO experiment, the magnetic field was applied with a polar angle of $\theta = 80^\circ$ and an azimuthal angle of $\varphi = 70^\circ$. This is the condition to obtain the maximum output signal, which is determined by the combination of spin-orbit torque (SOT) and anisotropic magnetoresistance. The former generates auto-oscillation and the latter determines output voltage. The SOT efficiency is largest for $\theta = 90^\circ$. However, when θ approaches 90° , coherent auto-oscillation is disturbed due to undesired magnetization switching. This is effectively suppressed by applying a small out-of-plane magnetic field [Appl. Phys. Lett. **111**, 032405 (2017)]. Therefore, we used $\theta = 80^\circ$, similar to the previous SHNO experiments [$\theta = 85^\circ$ in Phys. Rev. Lett. **114**, 137201 (2015), $\theta = 85^\circ$ in Phys. Rev. Appl. **11**, 064038 (2019)]. Moreover, a finite azimuthal angle of $\varphi = 70^\circ$ was chosen to achieve a large AMR signal. Note that similar azimuthal angles were used in previous works as well [$\varphi = 60^\circ$ in Phys. Rev. Lett. **114**, 137201 (2015), $\varphi = 60^\circ$ in Phys. Rev. Appl. **11**, 064038 (2019), $\varphi = 68^\circ$ in Sci. Adv. **5**, eaax8467 (2019), $\varphi = 68^\circ$ in Nat. Commun. **11**, 4006 (2020)].

Regarding the selection of the field angles, we have added the following sentence in our revised manuscript.

On page 8, “Here, we applied a magnetic field along the nearly in-plane direction to maximize the SOT-induced anti-damping effect. The slight tilt from the plane suppresses the SOT-induced magnetization switching [52]. The azimuthal angle was chosen to achieve a large electrical auto-oscillation signal [17,18].”

6. Looking at Fig.3(c-h), auto-oscillating signal dies off at higher fields for initial state and then starts to appear after applying positive gate voltage. However, there is no auto-

oscillations at lower magnetic fields when the gate voltage is -3V. Is it because of the irreversible changes in the gated region after applying positive gate voltage? If yes, can the original state be restored in these devices?

Response) We thank the reviewer for fruitful comment on the reversibility of the frequency tuning operation. As mentioned by the reviewer, the auto-oscillation signal is not clearly visible under certain field ranges in Figs. 3(c-h), where the intensity scale is displayed in the range from 1 dB to 3 dB. To clarify whether the auto-oscillation is not excited under those conditions, we rechecked each spectrum in Figs. 3(c-h). Figure R14 shows the auto-oscillation spectra for various magnetic fields ranging from 0.45 T to 0.60 T. Although there is a difference in the magnitude of the spectral intensity, it can be seen that auto-oscillation occurs in all magnetic field ranges. In particular, Fig. R14(a) showing the initial state is comparable to Fig. R14(e) for the state with $V_g = -3$ V. Therefore, to avoid possible misunderstanding, we have replotted the graphs in Figs. 3(c-h) with an intensity scale between 0.1 dB to 3.0 dB as shown in Figs. R15(a-e). Nevertheless, we do not clearly understand why the spectral intensities for $V_g = -3$ V are not fully restored to their initial states in all magnetic field regions, unlike the frequency. We speculate this is due to the stochastic nature of the VCMA effect based on ion migration as demonstrated in resistive memory devices, whose switching mechanism is the local movement of the oxygen ions [ACS Nano **7**, 2320-2325 (2013); IEEE Electron Device Lett. **36**, (2015)]. However, further investigation is required.

We have replotted the graphs in Figs. 3(c-h) with an intensity scale between 0.1 dB to 3.0 dB.

Figure R14 (a-e), Auto-oscillation spectra for magnetic fields varying between 0.45 T and 0.60 T with 0.05 T steps for sequentially applied gate voltages, $V_g = 0$ V (initial state) (a), $V_g = +4$ V (b), $V_g = +5$ V (c), $V_g = -2$ V (d), and $V_g = -3$ V (e). $I_{dc} = 2.9$ mA.

Figure R15 (a-e), Power spectral densities (PSDs) as a function of a magnetic field for sequentially applied gate voltages, $V_g = 0$ V (initial state) (a), $V_g = +4$ V (b), $V_g = +5$ V (c), $V_g = -2$ V (d), and $V_g = -3$ V (e). $I_{dc} = 2.9$ mA. Here, the PSD is scaled in the range from 0.1 dB to 3.0 dB.

7. In Fig. 4(a-e), slope of the frequency vs current is independent of gate voltage. Can the authors explain why the nonlinearity remains unaffected despite large changes in PMA? Authors may like to refer the following papers to address this. *Science Advances* 5 (9), eaax8467 [DOI: 10.1126/sciadv.eaax8467], *PHYSICAL REVIEW APPLIED* 9, 014017 (2018) [DOI: <https://doi.org/10.1103/PhysRevApplied.9.014017>]

Response) We appreciate the reviewer's critical comment on the nonlinearity of the oscillation frequency. According to the literatures mentioned by the reviewer [Sci. Adv. **5**, eaax8467 (2019) and Phys. Rev. Appl **9**, 014017 (2018)], the nonlinearity coefficient (N) of the auto-oscillation depends on the perpendicular magnetic anisotropy field (B_k). However, we did not observe a significant change in the current-dependent frequency, although the B_k changes by 0.24 T with the gate voltage. We attribute the weak nonlinearity to the large polar angle θ used for our measurements compared to the literature: $\theta = 10^\circ$ in the previous literature and $\theta = 80^\circ$ in our measurement. To verify this argument, we calculate the N value based on the equation used in the literature with our experimental parameters [Phys. Rev. B **97**, 184402 (2018)]. Figures R16(a) and R16(b) show the mapping of the N values as functions of B_k and external magnetic field (B_{ext}) with different polar angles (θ). The black lines represent the range of B_k change due to the VCMA effect extracted from Fig. 2 of the main text. When $\theta = 10^\circ$ [Figure R16(a)], the N varies largely with B_k , consistent with results in the literature [Science Advances **5**, eaax8467 (2019)]. On the other hand, for $\theta = 80^\circ$ (Fig. R16(b), our experimental condition), the change in N caused by the VCMA effect is considerably reduced compared to that for $\theta = 10^\circ$. Note that this result shows the trend of the non-linearity versus the measurement angle, but it cannot quantitatively explain the difference in the non-linearity between the two experiments. We believe that verification of the difference is beyond the scope of this manuscript and further investigation is required.

We have revised the manuscript as follows, and have added the above discussion in Supplementary Note 7. We have also added the suggested papers in reference of the revised manuscript (ref #55,57).

On page 10, "Note that the weak dependence of the slope of oscillation frequency versus the I_{dc} curves on V_g is different from the previous results [53,55,56]. This might be due to the different polar angles used for the measurements (Supplementary Note 6)."

Figure R16 (a,b), Calculated nonlinearity coefficient (N) as functions of perpendicular magnetic anisotropy field (B_k) and external magnetic field (B) with a different polar angles (θ) of magnetic field, $\theta = 10^\circ$ (a), $\theta = 80^\circ$ (b). The black bar in each figure represents the area of B_k change by VCMA ($B_k = -0.08 \text{ T} \sim 0.16 \text{ T}$) at the magnetic field where the auto-oscillation spectra were measured.

8. There are few typos in the manuscript and I recommend authors to carefully proof-read the manuscript to improve the readability of the paper for a general audience. For ex- in page no 5 and 12, the unit after 150 seems missing.

Response) We apologize for the typos and missing unit. We have inserted the Celsius unit on page 5 and page 11 of the revised manuscript. Moreover, the manuscript has been thoroughly proofread by a professional editor.

Reviewers' Comments:

Reviewer #1:

Remarks to the Author:

The authors have made genuine efforts to address some of the issues raised by both referees and they have provided in many places satisfactory responses. In particular, they added the additional experiments and necessary details. However, some new concerns arise according to the authors' response, which must be clarified or corrected.

1. There is missing the size information of the test device in Note 2 in Supplementary Information.
2. The authors make the statement as "Note that the out-of-plane magnetic field geometry was employed to accurately determine the PMA, avoiding artifacts that might arise in the in-plane magnetic field geometry, such as local anisotropy variation or two magnon scattering". As far as I know, two magnon scattering contributes to the linewidth rather than the resonance field for in-plane magnetization. How does two magnon scattering disturb the accurate determination of the PMA? I guess that this statement about two magnon scattering is out of place.
3. For additional resistance and f_{auto} variations by gate voltages in Note 4, what is the temperature during gating voltage V_g ? Is it at 150 C as the statement in the main text?

Reviewer #3:

Remarks to the Author:

I appreciate the authors for responding to comments in a constructive and professional way. While the authors have thoroughly addressed the comments of all reviewers, I am not convinced with their response regarding the nature of the auto-oscillating mode in their devices.

In their response to Rev#1Q5, the authors claim non-propagating auto-oscillating mode in their devices and they have provided a Fig. R5 to support their argument. From the Figure, it is clear that AO frequency stays below the FMR for different PMA fields but I wonder what is the physical mechanism for such localized modes. According to the non-linear theory of magnetodynamics, a negative nonlinearity causes the auto-oscillation frequency decrease with amplitude, pushing it into the spin-wave bandgap, where it first self-localizes, and further nucleates into magnetodynamical solitons such as spin-wave bullets in in-plane magnetic thin films or droplets in films with large PMA. However, in the present work, I notice AO mode starts well below FMR and the non-linearity stays positive. Can authors clarify the physical mechanism to explain this behavior? Before I recommend the manuscript for publication, I suggest authors should clarify this critical issue by repeating the same measurement with carefully measuring the FMR frequency and AO frequency under identical conditions.

Reviewer #1 (Remarks to the Author)

The authors have made genuine efforts to address some of the issues raised by both referees and they have provided in many places satisfactory responses. In particular, they added the additional experiments and necessary details. However, some new concerns arise according to the authors' response, which must be clarified or corrected.

Response) We appreciate the reviewer's comment that "*The authors have made genuine efforts to address some of the issues raised by both referees and they have provided in many places satisfactory responses. In particular, they added the additional experiments and necessary details.*" We respond to the reviewer's additional comments below, which hopefully alleviates the reviewer's concern so that the revised manuscript is now acceptable for publication.

1. There is missing the size information of the test device in Note 2 in Supplementary Information.

Response) We have inserted the dimension of sample size for fast operation of voltage-controlled magnetic anisotropy in Supplementary Note 2.

On page 3 in Supplementary Note 2, "*The sample was patterned into a Hall bar device with a $10\ \mu\text{m} \times 10\ \mu\text{m}$ Hall cross, which is the same structure as the VCMA test sample shown in Fig. 1c of the main text.*"

2. The authors make the statement as "Note that the out-of-plane magnetic field geometry was employed to accurately determine the PMA, avoiding artifacts that might arise in the in-plane magnetic field geometry, such as local anisotropy variation or two magnon scattering". As far as I know, two magnon scattering contributes to the linewidth rather than the resonance field for in-plane magnetization. How does two magnon scattering disturb the accurate determination of the PMA? I guess that this statement about two magnon scattering is out of place.

Response) We thank the reviewer for this comment. It has been reported in previous literatures [Phys. Rev. B **80** 180415 (2009), IEEE Trans. Magn. **40**, 2-11 (2004)] that two-magnon scattering (TMS) can modify the resonance field as well as the linewidth of ST-FMR. TMS occurs when a spin wave dispersion with finite- k modes are degenerated with a $k = 0$ mode. This unique dispersion condition can be observed when a magnetic field is applied along the in-plane direction [see Fig. 4 of Phys. Rev. B **80** 180415 (2009)]. Through the TMS, disorders

can couple the uniform precessional mode ($k = 0$) to finite- k spin wave modes ($k \neq 0$). These spin waves with finite- k induce an effective magnetic field, resulting in a shift of the resonance field (or resonance frequency) [section IV of IEEE Trans. Magn. **40**, 2-11 (2004)]. To confirm that the TMS occurs in our sample, we conducted the ST-FMR measurement with different polar angles of magnetic fields (θ) in Ta (3 nm)/Pt (5 nm)/[Co (0.45 nm)/Ni(0.6 nm)]/Co (0.45 nm)/AlO_x (2 nm). Note that all measurement conditions except for θ are identical to those for the measurement shown in Fig. 2 of the main text. Figure R1(a) shows the normalized ST-FMR spectra for different polar angles between $\theta = 5^\circ$ and $\theta = 90^\circ$ with 5° steps. Here, we fixed the measurement frequency of 7 GHz. The extracted resonance field (B_{res}) from Fig. R1(a) was plotted as a function of θ in Fig. R1(b). The solid line in Fig. R1(b) is the fitting curve using the angular dependent Kittel equation [Eq.6 of Phys. Rev. B **90**, 224428 (2014)]. As θ increases, the B_{res} deviates from the Kittel equation, which is partly attributed to the TMS that occurs when a magnetic field is applied near an in-plane direction [Fig. 1(b) of Phys. Rev. B **80**, 180415 (2009)]. This experiment indicates that the TMS is present in our sample, which can cause a resonance field shift under a near-in-plane magnetic field.

We have added the above discussion in Supplementary Note 3.

Figure R1 (a) Normalized ST-FMR spectra at 7 GHz depending on the polar angle (θ) of the magnetic field varying between 5° and 90° with 5° steps. **c**, Resonance field (B_{res}) as a function of θ extracted from Fig. R1(a). The solid line is a fit to the experimental data point using the Kittel equation associated with magnetic anisotropy.

3. For additional resistance and f_{auto} variations by gate voltages in Note 4, what is the temperature during gating voltage V_g ? Is it at 150 C as the statement in the main text?

Response) We apologize for the missing information. The experiments shown in Supplementary Note 4 were performed at room temperature, identical to those in Fig.5 of the main text. We have modified the following sentence in Supplementary Note 5 (Supplementary Note 4 of the previous version).

On page 6 in Supplementary Note 5, *“To confirm this, we repeated the same measurement shown in Fig. 5 of the main text with a dc current (I_{dc}) of 2.1 mA and a voltage (V_g) duration time of 30 s at room temperature.”*

Reviewer #3 (Remarks to the Author)

I appreciate the authors for responding to comments in a constructive and professional way. While the authors have thoroughly addressed the comments of all reviewers, I am not convinced with their response regarding the nature of the auto-oscillating mode in their devices.

In their response to Rev#1Q5, the authors claim non-propagating auto-oscillating mode in their devices and they have provided a Fig. R5 to support their argument. From the Figure, it is clear that AO frequency stays below the FMR for different PMA fields but I wonder what is the physical mechanism for such localized modes. According to the non-linear theory of magnetodynamics, a negative nonlinearity causes the auto-oscillation frequency decrease with amplitude, pushing it into the spin-wave bandgap, where it first self-localizes, and further nucleates into magnetodynamical solitons such as spin-wave bullets in in-plane magnetic thin films or droplets in films with large PMA. However, in the present work, I notice AO mode starts well below FMR and the non-linearity stays positive. Can authors clarify the physical mechanism to explain this behavior? Before I recommend the manuscript for publication, I suggest authors should clarify this critical issue by repeating the same measurement with carefully measuring the FMR frequency and AO frequency under identical conditions.

Response) We appreciate the reviewer's comment that "*I appreciate the authors for responding to comments in a constructive and professional way*". On the other hand, she/he is concerned about the spin wave mode of auto-oscillation in our spin Hall nano oscillator (SHNO). We below respond to the reviewer's comments with additional experiments, which hopefully alleviates the reviewer's concern so that the revised manuscript is now acceptable for publication.

In our previous response, we compared the auto-oscillation frequency with the FMR frequency *calculated* from the anisotropy value and concluded that the auto-oscillation in our device occurs in a localized mode, because the auto-oscillation frequencies lie below the *calculated* FMR frequency. However, this contradicts the positive nonlinearity, as the reviewer pointed out. To clarify this problem, we followed the reviewer's suggestion to compare the FMR and auto oscillation frequencies measured using the same sample with identical conditions.

We first measured auto-oscillation power spectral density using a 100 nm-constriction SHNO device of a Ta (3 nm)/Pt (5 nm)/[Co (0.45 nm)/Ni (0.6 nm)]₇/Co (0.45 nm)/AlO_x (2 nm) structure. Figure R2(a) shows the auto-oscillation power spectral density as a function of the

magnetic field at a dc current of 2.3 mA. Here, the polar (θ) and the azimuthal angles (φ) of the applied magnetic field are 80° and 70° , respectively. We also performed the ST-FMR measurement using the same device under the same magnetic field direction. Note that the ST-FMR measurement was done with a microwave power of -17 dBm, but no dc current is applied. Figure R2(b) shows the resonance frequency (f_{res}) of the ST-FMR as a function of the resonance field (B_{res}). Notably, the auto-oscillation frequency is higher than the ST-FMR f_{res} for the same magnetic field, which is opposite to those previously obtained from the calculations.

We further confirm this by measuring the ST-FMR while increasing a dc current. Figure R2(c) shows the results. For small currents ($0 \sim 1.9$ mA), only one resonance peak appears near ~ 0.70 T, which corresponds to the FMR resonance field. For large current (~ 2.3 mA), on the other hand, another peak appears near ~ 0.35 T, which corresponds to the auto-oscillation. These results corroborate that the auto-oscillation frequency is higher than the FMR frequency. Therefore, the auto-oscillation in our device occurs in a propagation mode. This is consistent with a positive nonlinearity of our SHNO device.

We again appreciate the reviewer for her/his advice to help us to correct our mistake. We have revised Supplementary Note 9 to include the above discussion and modified the following sentence in the main text.

On page 10, “*We also note that the auto-oscillation dynamics in our samples belongs to the propagation mode because the auto-oscillation frequency is larger than the FMR frequency [53,57,58] (Supplementary Note 9).*”

Figure R2 (a) The auto-oscillation power spectral density as a function of magnetic field at a dc current of 2.3 mA for a 100 nm-constriction SHNO device of a Ta (3 nm)/Pt (5 nm)/[Co (0.45 nm)/Ni (0.6 nm)]₇/Co (0.45 nm)/AlO_x (2 nm) structure. (b) Resonance frequency (f_{res}) as a function of resonance field (B_{res}) without dc current. (c) ST-FMR spectra for three different dc currents of 0, 1.9 mA, and 2.3 mA. $\theta = 80^\circ$ and $\varphi = 70^\circ$.

Reviewers' Comments:

Reviewer #1:

Remarks to the Author:

The answer of the authors to my comments is satisfactory. Therefore I recommend publication.

Reviewer #3:

Remarks to the Author:

The authors have addressed my comments. In my opinion, the manuscript can be published in Nature Communications without any additional revisions.